# Multi-Fidelity Active Learning with GFlowNets

**Alex Hernandez-Garcia**[*]    *alex.hernandez-garcia@mila.quebec*
*Mila, Université de Montréal*

**Nikita Saxena**[*][†]    *nikitasaxena0209@gmail.com*
*Birla Institute of Technology and Science, Pilani*

**Moksh Jain**    *moksh.jain@mila.quebec*
*Mila, Université de Montréal*

**Cheng-Hao Liu**    *chenghao.liu@mail.mcgill.ca*
*Mila, McGill University*

**Yoshua Bengio**    *yoshua.bengio@mila.quebec*
*Mila, Université de Montréal, CIFAR Fellow, IVADO*

**Reviewed on OpenReview:** *https://openreview.net/forum?id=dLaazW9zuF*

## Abstract

In the last decades, the capacity to generate large amounts of data in science and engineering applications has been growing steadily. Meanwhile, machine learning has progressed to become a suitable tool to process and utilise the available data. Nonetheless, many relevant scientific and engineering problems present challenges where current machine learning methods cannot yet efficiently leverage the available data and resources. For example, in scientific discovery, we are often faced with the problem of exploring very large, structured and high-dimensional spaces. Moreover, the high fidelity, black-box objective function is often very expensive to evaluate. Progress in machine learning methods that can efficiently tackle such challenges would help accelerate currently crucial areas such as drug and materials discovery. In this paper, we propose a multi-fidelity active learning algorithm with GFlowNets as a sampler, to efficiently discover diverse, high-scoring candidates where multiple approximations of the black-box function are available at lower fidelity and cost. Our evaluation on molecular discovery tasks shows that multi-fidelity active learning with GFlowNets can discover high-scoring candidates at a fraction of the budget of its single-fidelity counterpart while maintaining diversity, unlike RL-based alternatives. These results open new avenues for multi-fidelity active learning to accelerate scientific discovery and engineering design.

## 1 Introduction

To tackle the most pressing challenges for humanity, such as the climate crisis and the threat of pandemics or antibiotic resistance, there is a growing need for new scientific discoveries. By way of illustration, new materials can play an important role in improving the efficiency of energy production and storage; and reducing the costs and duration of drug discovery cycles has the potential to effectively and rapidly mitigate

---

[*]Equivalent Contribution
[†]Work done during an internship at Mila

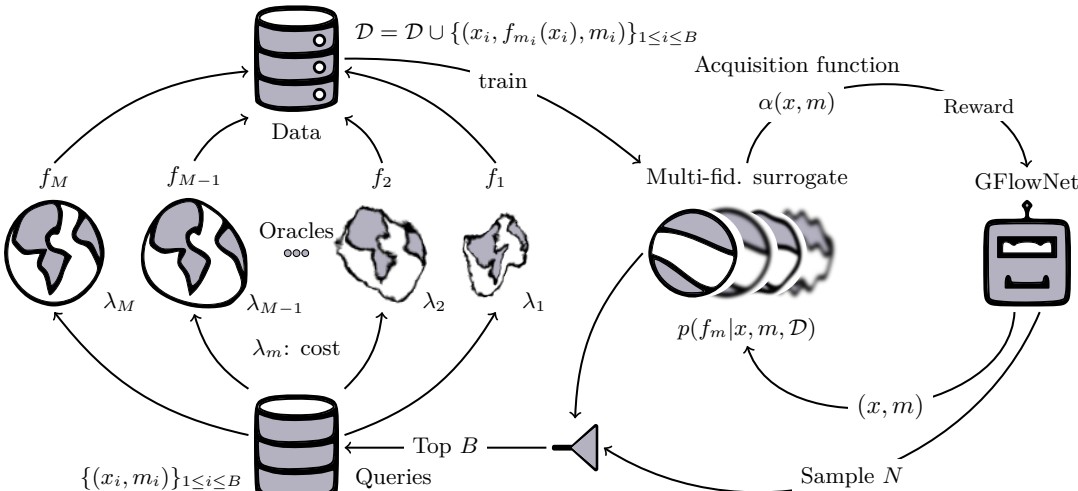

Figure 1: Illustration of multi-fidelity active learning with GFlowNets (Algorithm 1). Given a set of $M$ oracles $f_1, \ldots, f_M$ (center left) with varying fidelities and costs $\lambda < \ldots < \lambda_M$, respectively, we can construct a data set $\mathcal{D}$ (top left) with annotations from the oracles. With this data, we fit a multi-fidelity surrogate (center), modelling the posterior $p(f_m(x)|x, m, \mathcal{D})$. The surrogate is used to evaluate a multi-fidelity acquisition function—max-value entropy search in our experiments— which makes the reward to train a GFlowNet (right). The GFlowNet is trained to sample both an object $x$ and the fidelity $m$ proportionally to this reward. Once the GFlowNet is trained, we sample $N$ tuples $(x, m)$ and select the top $B$ according to the acquisition function (bottom left). Finally, we annotate each candidate with the selected oracle, add them to the data set and repeat the process until the budget is exhausted.

the consequences of new diseases. In recent years, researchers in materials science, biochemistry and other fields have increasingly adopted machine learning as a tool since it holds the promise to drastically accelerate scientific discovery (Butler et al., 2018; Zitnick et al., 2020; Bashir et al., 2021; Das et al., 2021).

Although machine learning has already made a positive impact in scientific discovery applications (Stokes et al., 2020; Jumper et al., 2021), unleashing its full potential demands improving the current algorithms (Agrawal & Choudhary, 2016). For example, typical tasks in potentially impactful applications in materials and drug discovery require exploring combinatorially large, structured and high-dimensional spaces (Bohacek et al., 1996; Polishchuk et al., 2013), where only small, noisy data sets are available. Furthermore, obtaining new annotations computationally or experimentally is often very expensive. Such scenarios present serious challenges even for the most advanced machine learning methods currently available.

In the search for a useful discovery, we typically define a quantitative proxy for usefulness, which we can view as a black-box function. One promising avenue for improvement is developing methods that more efficiently leverage the availability of multiple approximations of the target black-box function at lower fidelity but much lower cost than the highest fidelity oracle (Chen et al., 2021; Fare et al., 2022). For example, a standard tool to characterise the properties of materials and molecules is quantum mechanics simulations such as Density Functional Theory (DFT) (Parr, 1980; Sholl & Steckel, 2022). However, DFT is computationally too expensive for high-throughput exploration of large search spaces. Thus, large-scale exploration can only be achieved through cheaper but less accurate oracles. Nonetheless, solely relying on low-fidelity approximations is clearly suboptimal. Ideally, such "needle-in-a-haystack" problems would be best tackled by methods that can efficiently and adaptively distribute the available computational budget between the multiple oracles depending on the already acquired information.

Another challenge is that even the highest fidelity oracles are often underspecified with respect to the actual, relevant, downstream applications. This underspecification problem can be mitigated by finding multiple candidate solutions (Jain et al., 2023a). However, most current machine learning methods used

in scientific discovery problems, such as Bayesian optimisation (BO, Song et al., 2018; Garnett, 2023) and reinforcement learning (RL, Angermueller et al., 2020), are designed for global optimisation of the target function. Therefore, it is imperative to develop methods that not only find the global optimum, but also discover sets of diverse, high-scoring candidates.

Recently, generative flow networks (GFlowNets or GFN, Bengio et al., 2021a) have demonstrated their ability to find diverse candidates through discrete probabilistic modelling, with particularly promising results when used in an active learning loop (Jain et al., 2022). In this paper, we propose a multi-fidelity active learning algorithm enhanced with these capabilities of GFlowNets. Our contributions can be summarized as follows:

- We introduce a multi-fidelity active learning algorithm designed for combinatorially large, structured and high-dimensional spaces.

- We propose an extension of GFlowNets for this multi-fidelity setting, to sample both candidates and oracle indices, proportionally to a given acquisition function.

- We conduct a comprehensive empirical evaluation across four scientific discovery tasks and demonstrate that multi-fidelity active learning with GFlowNets

  - discovers high-scoring samples with reduced computational costs compared to its single-fidelity counterpart, and
  - identifies multiple modes of the target function, unlike methods relying on reinforcement learning or standard Bayesian optimisation, thereby facilitating diverse sampling.

## 2 Related Work

Our work can be framed within the broad field of active learning (AL), a class of machine learning methods whose goal is to learn an efficient data sampling scheme to accelerate training (Settles, 2009). For the bulk of the literature in AL, the goal is to train an accurate model $h(x)$ of an unknown target function $f(x)$, as in classical supervised learning. However, in certain scientific discovery problems, which motivate our work, a desirable goal is often instead to discover multiple, diverse candidates $x$ with high values of $f(x)$, as discussed in Section 1.

Our work is also closely connected to Bayesian optimisation (BO, Garnett, 2023; Snoek et al., 2015), which aims at optimising a black-box objective function $f(x)$ that is expensive to evaluate. In contrast to the problems we address in this paper, standard BO typically considers continuous domains and works best in relatively low-dimensional spaces (Frazier, 2018). Nonetheless, in recent years, approaches for BO with structured data (Deshwal & Doppa, 2021; Papenmeier et al., 2023) and high-dimensional domains (Grosnit et al., 2021) have been proposed in the literature. The main difference between standard BO and the problem we tackle in this paper is that we are interested in finding multiple, diverse samples with high value of $f$ and not only the optimum. Recent work by Maus et al. (2022) has proposed a variant of traditional BO to find diverse solutions.

This goal, as well as the discrete nature of the search space, is shared with active search (Garnett et al., 2012), a variant of active learning in which the task is to efficiently find multiple samples of a valuable (binary) class from a discrete domain $\mathcal{X}$. This objective was already considered in the early 2000s by Warmuth et al. (2001) for drug discovery, and more formally analysed in later work (Jiang et al., 2017; 2019). Another recent research area in stochastic optimisation that considers diversity is so-called Quality-Diversity (Chatzilygeroudis et al., 2021), which typically uses evolutionary algorithms that search in a latent space. These and other problems such as multi-armed bandits (Robbins, 1952) and the general framework of experimental design (Chaloner & Verdinelli, 1995) all share the objective of optimising or exploring an expensive black-box function. Formal connections between some of these areas have been established in the literature (Srinivas et al., 2010; Foster, 2021; Jain et al., 2023a; Fiore et al., 2023).

Multi-fidelity methods have been proposed in most of these areas of research. An early survey on multi-fidelity methods for Bayesian optimisation was compiled by Peherstorfer et al. (2018), and research on the subject has continued since with the proposal of specific acquisition functions (Takeno et al., 2020) and the

use of deep neural networks to improve the modelling (Li et al., 2020). Recently, works on multi-fidelity active search have also appeared in the literature (Nguyen et al., 2021), but interestingly, the literature on multi-fidelity active learning (Li et al., 2022a) is scarcer. Finally, while multi-fidelity methods have started to be applied in scientific discovery problems (Chen et al., 2021; Fare et al., 2022) the literature is still limited probably because most approaches cannot tackle the specifics of scientific discovery, such as the need for diverse samples. Here, we aim at addressing this need with the use of GFlowNets (Bengio et al., 2021a; Jain et al., 2023b) for multi-fidelity active learning.

## 3 Method

In this section, we first briefly introduce the necessary background on GFlowNets and active learning. Then, we describe the proposed algorithm for multi-fidelity active learning with GFlowNets.

### 3.1 Background

**GFlowNets**  Generative flow networks (GFN; Bengio et al., 2021a;b) are amortised samplers originally designed for sampling from discrete high-dimensional distributions. Given a space of compositional objects $\mathcal{X}$ and a non-negative reward function $R(x)$, GFlowNets are designed to learn a stochastic policy $\pi(x)$ that generates $x \in \mathcal{X}$ with a probability proportional to the reward, that is $\pi(x) \propto R(x)$. This distinctive property induces sampling of diverse, high-reward objects, which is a desirable property for scientific discovery, among other applications (Jain et al., 2023a).

A key property of GFlowNets is that objects $x \in \mathcal{X}$ are constructed sequentially by sampling transitions $s_t \rightarrow s_{t+1} \in \mathbb{A}$ between partially constructed objects (states) $s \in \mathcal{S}$, which includes a unique empty state $s_0$. The stochastic forward policy is typically parameterised by a neural network $P_F(s_{t+1}|s_t; \theta)$, where $\theta$ denotes the learnable parameters, which models the distribution over transitions from the current state $s_t$ to the next state $s_{t+1}$. The backward transitions are parameterised too and denoted $P_B(s_t|s_{t+1}; \theta)$. Objects $x$ are generated by the sequential application of $P_F$, forming trajectories $\tau = (s_0 \rightarrow s_1 \ldots \rightarrow x)$. To learn the parameters $\theta$ such that $\pi(x) \propto R(x)$ we use the trajectory balance learning objective (Malkin et al., 2022)

$$\mathcal{L}_{TB}(\tau; \theta) = \left( \log \frac{Z_\theta \prod_{t=0}^n P_F(s_{t+1}|s_t; \theta)}{R(x) \prod_{t=1}^n P_B(s_t|s_{t+1}; \theta)} \right)^2, \tag{1}$$

where $Z_\theta$ is a trainable approximation of the partition function $\sum_{x \in \mathcal{X}} R(x)$. The GFlowNet learning objective supports training from off-policy trajectories, so during training the trajectories are typically sampled from a mixture of the current policy with a uniform random policy. The reward can also be tempered to make the policy focus on the modes (see Appendix C.3).

**Active Learning**  In its simplest formulation, the (single fidelity) active learning problem that we consider is as follows: we start with an initial data set $\mathcal{D} = \{(x_i, f(x_i))\}$ of samples $x \in \mathcal{X}$ and their evaluations by an expensive, black-box objective function (oracle) $f : \mathcal{X} \rightarrow \mathbb{R}$, which we use to train a surrogate model $h(x)$. A GFlowNet can then be trained to learn a generative policy $\pi_\theta(x)$ using $h(x)$ as reward function, that is $R(x) = h(x)$. After training, the policy $\pi_\theta(x)$ can be used to generate a batch of samples to be evaluated by the oracle $f$, add them to the data set and repeat the process a number of active learning rounds.

As an alternative, instead of directly using the surrogate output as the reward, we can instead train a probabilistic surrogate $p(f|\mathcal{D})$ and use as reward the output of an acquisition function $\alpha(x, p(f|\mathcal{D}))$ that considers the epistemic uncertainty of the surrogate model, as typically done in Bayesian optimisation. This is the approach by Jain et al. (2022) with GFlowNet-AL. An important difference between traditional BO and active learning with GFlowNets is that the latter samples from the acquisition function instead of optimising it (Jain et al., 2023a). This difference accounts for the capability of active learning with GFlowNets to discover diverse candidates: if multiple, diverse candidates have high values of the acquisition function, GFlowNet has the potential to generate them with high probability, whereas standard BO seeks to find the optimum only.

While much of the active learning literature (Settles, 2009) has focused on so-called *pool-based* active learning, where the learner selects samples from a pool of unlabelled data, we here consider the scenario of *de novo query synthesis*, where samples are selected from the entire object space $\mathcal{X}$. This scenario is particularly suited for scientific discovery (King et al., 2004; Xue et al., 2016; Yuan et al., 2018; Kusne et al., 2020). The ultimate goal pursued in active learning applications is also heterogeneous. Often, the goal is the same as in classical supervised machine learning: to train an accurate (surrogate) model $h(x)$ of the unknown target function $f(x)$. In many scientific discovery problems, we are not interested in the surrogate's accuracy across the entire input space $\mathcal{X}$, but rather in discovering new, diverse objects with high values of $f$. We have reviewed the literature that is connected to our work in Section 2.

## 3.2 Multi-Fidelity Active Learning

We now consider the following active learning problem with multiple oracles of different fidelities. Our ultimate goal is to generate a batch of $K$ samples $\{x_i\}_{i=1}^K \in \mathcal{X}$ according to the following desiderata:

- The samples obtain a high value when evaluated by the objective function $f : \mathcal{X} \to \mathbb{R}^+$.

- The samples are diverse, covering distinct, high-valued regions of $f$.

Furthermore, we are constrained by a computational budget $\Lambda$ that limits our capacity to evaluate $f$. While $f$ is extremely expensive to evaluate, we have access to a discrete set of approximate functions (oracles) $\{f_m\}_{1 \le m \le M} : \mathcal{X} \to \mathbb{R}^+$, where $m$ represents the fidelity index and each oracle has an associated cost $\lambda_m$ and level of confidence $\ell_m \in (0, 1]$. We assume, without loss of generality, that the larger $m$, the higher the fidelity or confidence, that $\lambda_1 < \lambda_2 < \ldots < \lambda_M < \Lambda$ and that $\ell_M = 1$. We also assume $f_M = f$ because, even though there may exist more accurate oracles, we do not have access to them. This scenario resembles many practically relevant problems in scientific discovery and motivates our approach: because the objective function $f_M$ is not a perfect proxy of the true usefulness of objects $x$, we seek diversity; and because $f_M$ may be expensive to evaluate, we make use of approximate models.

In multi-fidelity active learning—as well as in multi-fidelity Bayesian optimisation—the iterative sampling scheme consists of not only selecting the next object $x$ (or batch of objects) to evaluate, but also the level of fidelity $m$, such that the procedure is cost-effective. Namely, after each active learning round we acquire triplets $(x_i, f_{m_i}(x_i), m_i)$.

Briefly, our algorithm, MF-GFN follows these iterative steps: at each iteration $j$, we use the currently available data $\mathcal{D}_j$ to train a probabilistic multi-fidelity surrogate model $h(x, m)$. We can use the surrogate to compute the worth of annotating a candidate $x$ with the oracle $f_m$ via an acquisition function $\alpha(x, m)$. Next, we train a GFlowNet with the acquisition function as a reward. Once trained, we sample $N$ tuples $(x, m)$ and select the top $B$, as per the acquisition function. Finally, we annotate each candidate $x$ with the selected oracle $m$ and start over with the extended data set. Figure 1 contains a visual illustration of MF-GFN and more detailed descriptions are provided in Algorithm 1 and in Appendices A and B. Below, we further describe the surrogate model and the acquisition function, and in Section 3.3 we introduce multi-fidelity GFlowNets.

**Surrogate Model**   Given a dataset $\mathcal{D}$, a candidate $x$ and an oracle index $m$, we want to model the posterior distribution over the output of the oracle, $p(f_m(x)|x, m, \mathcal{D})$. A natural modelling choice is Gaussian Processes (GP, Rasmussen & Williams, 2005), commonly used in Bayesian optimisation. However, in order to better model structured, high-dimensional data, we use deep kernel learning (Wilson et al., 2016): First, a non-linear embedding of the inputs $z = g_\omega(\mathrm{x})$ is learnt by a deep neural network with parameters $\omega$. Then, we use the following multi-fidelity GP kernel:

$$K_{MF}((x, m), (x', m')) = K_X(g_\omega(x), g_\omega(x')) + K_M(m, m') \times K_{X_M}(g_\omega(x), g_\omega(x')), \tag{2}$$

where both $K_X(z, z')$ and $K_{X_M}(z, z')$ are Matérn kernels (with a different lengthscale each) applied to the latent representations of the inputs from the network and

$$K_M(m, m') = (1 - \ell_m)(1 - \ell_{m'})(1 + \ell_m \ell_{m'})^p$$

---

**Algorithm 1:** MF-GFN: Multi-fidelity active learning with GFlowNets. A graphical summary of this algorithm is shown in Fig. 1.

---

**Input:** $\{(f_m, \lambda_m, \ell_m)\}$: $M$ oracles and their corresponding cost and confidence;
$\mathcal{D}_0 = \{(x_i, f_{m_i}(x_i), m_i)\}$: Initial data set;
$h(x, m)$: Multi-fidelity Gaussian Process surrogate model;
$\alpha(x, m)$: Multi-fidelity acquisition function;
$R(\alpha(x, m), \beta)$: reward function with temperature parameter $\beta$ to train the GFlowNet;
$N$: Number of candidates sampled from the GFlowNet;
$B$: Acquisition batch size of oracles queries;
$\Lambda$: Maximum available budget;
$K$: Number of top-scoring candidates to be evaluated at termination;
**Initialisation:** $\mathcal{D} \leftarrow \mathcal{D}_0$, $\Lambda_j \leftarrow 0$, $j \leftarrow 0$
**while** $\Lambda_j < \Lambda$ **do**
- Fit surrogate $h$ on data set $\mathcal{D}$;
- Train GFlowNet with reward $R(\alpha(x, m), \beta)$ to obtain policy $\pi_\theta(x)$;
- Sample $N \gg B$ tuples $(x_i, m_i) \sim \pi_\theta$;
- Score each tuple using $\alpha(x, m)$ and select the top $B$ tuples with the highest scores;
- Evaluate each tuple with the corresponding oracle to form batch
  $\mathcal{B} \leftarrow \{(x_1, f_{m_1}(x_1), m_1), \ldots, (x_B, f_{m_B}(x_B), m_B)\}$;
- Update data set $\mathcal{D} \leftarrow \mathcal{D} \cup \mathcal{B}$, budget $\Lambda_j \leftarrow \Lambda_j + \sum_{i=1}^{B} \lambda_{m_i}$ and index $j \leftarrow j + 1$;

**end**
**Result:** Top-$K(\mathcal{D})$, set of K candidates with largest values of the highest-fidelity oracle $f_M$.

---

is applied over the fidelity confidences, which we set equal to the normalised costs $\ell_m = \frac{\lambda_m}{\lambda_M} \in (0, 1]$ for simplicity. The kernel $K_{MF}$ is known as linear truncated kernel, is implemented in BoTorch (Balandat et al., 2020) and has been used before in multi-fidelity Bayesian optimisation works, for example by Mikkola et al. (2023). Additional details are provided in Appendix B. Finally, we would like to note that our multi-fidelity active learning algorithm is independent of the specific choice of surrogate model.

**Acquisition Function** Multi-fidelity methods proposed in the Bayesian optimisation literature have adapted information theory-based acquisition functions (Li et al., 2022a; Wu et al., 2023; Li et al., 2022b). In our work, we use the multi-fidelity version (Takeno et al., 2020) of max-value entropy search (MES, Wang & Jegelka, 2017). MES captures the mutual information between the value of candidate $x$ and the maximum value attained by the objective function, $f^\star$. The multi-fidelity variant, MF-MES, is designed to select the candidate $x$ and the fidelity $m$ that maximise the mutual information between $f_M^\star$ and the oracle at fidelity $m$, $f_m$, weighted by the cost of the oracle $\lambda_m$:

$$\alpha(x, m) = \frac{1}{\lambda_m} I(f_M^\star; f_m(x)|\mathcal{D}_j). \tag{3}$$

We chose MES as the acquisition function for our approach because it has been shown to be more efficient than plain entropy search (Wang & Jegelka, 2017). However, the algorithm is independent of the choice of acquisition function. Additionally, for efficiency, we adopt the GIBBON approximation of MF-MES, which has demonstrated good performance in the context of multi-fidelity optimisation (Moss et al., 2021). We provide further details about the acquisition function and the GIBBON approximation in Appendix B.

### 3.3 Multi-Fidelity GFlowNets

A multi-fidelity acquisition function can be regarded as a cost-adjusted utility function. Therefore, in order to carry out a cost-aware search, we seek to sample diverse objects with high value of the acquisition function. To this purpose, we propose to use a GFlowNet as a generative model by training it to sample the fidelity $m$ in addition to the candidate $x$ itself. In other words, it learns a sampling policy $\pi_\theta(x, m)$. Formally, given a GFlowNet with state and transition spaces $\mathcal{S}$ and $\mathbb{A}$, we augment the state space with a new dimension

for the fidelity $\mathcal{M}' = \{0, 1, 2, \ldots, M\}$ (including $m = 0$, which corresponds to unset fidelity), such that the augmented, multi-fidelity state space is $\mathcal{S}_M = \mathcal{S} \times \mathcal{M}'$. The set of allowed transitions $\mathbb{A}_M$ is augmented such that a fidelity $m > 0$ of a trajectory must be selected once, and only once, from any intermediate state. Intuitively, allowing the selection of the fidelity at any step in the trajectory should give flexibility for better generalisation. At the end, complete trajectories are the concatenation of an object $x$ and the fidelity $m > 0$, that is $(x, m) \in \mathcal{X}_M = \mathcal{X} \times \mathcal{M}$.

In summary, the proposed approach learns a policy that jointly samples objects in a possibly very large, structured and high-dimensional space, together with the level of fidelity. Since we use a multi-fidelity acquisition function as reward function to train the multi-fidelity GFlowNet, the learnt sampling policy will be approximately proportional to the acquisition function, hence providing diversity. As in the single-fidelity case, if multiple, diverse tuples $(x, m)$ obtain high values of the acquisition function $\alpha$, the GFlowNet can potentially sample them with high probability. In practice, we rescale the acquisition function so as to further increase the relative rewards of high values of $\alpha$, as detailed in Appendix C.3.

## 4 Empirical Evaluation

In this section, we present empirical evaluations of multi-fidelity active learning with GFlowNets. Through our experiments, we aim to answer the following questions:

- Can our multi-fidelity active learning approach find high-scoring, diverse samples at lower cost than with a single-fidelity oracle?

- Does MF-GFN, which samples objects and fidelities $(x, m)$, provide any advantage over sampling only $x$ and selecting $m$ randomly?

### 4.1 Metrics

As discussed in Section 3.2, our goal is to sample diverse objects with high scores according to a reward function. Following Gao et al. (2022) and Jain et al. (2022), we here consider a pair of metrics that capture both the scores and the diversity of the final batch of candidates. These metrics are computed from the set of best $K$ candidates, obtained out of the full data set $\mathcal{D}_j$ at iteration $j$. We refer to this set as top-$K(\mathcal{D}_j)$, which consists of the $K$ candidates with the largest values of the objective function (highest-fidelity oracle) $f_M$. The two metrics we consider in this section are:

- **Mean top-K score**: mean score, per the highest fidelity oracle $f_M$, of the top-$K$ samples.

- **Top-K diversity**: mean pairwise distance within the top-$K$ samples.

Additional details and formal definitions are provided in Appendix D. We report the metrics averaged over 3 experiments with different random seeds, as well as the 95 % confidence intervals of the mean top-$K$ scores estimated via bootstrapping.

Since here we are interested in the cost effectiveness of the active learning process, we evaluate the above metrics as a function of the cost accumulated in querying the oracles. It is important to note that multi-fidelity approaches are *not* aimed at achieving *better* mean top-$K$ scores than a single-fidelity active learning counterpart, but rather *the same* mean top-$K$ scores but *with a smaller budget*.

### 4.2 Baselines

In order to evaluate our approach, and to shed light on the questions stated above, we consider the following baselines:

**GFlowNet with highest fidelity (SF-GFN)** GFlowNet-based active learning (GFlowNet-AL) as in Jain et al. (2022) with the highest fidelity oracle, to establish a benchmark for performance without considering the cost-accuracy trade-offs.

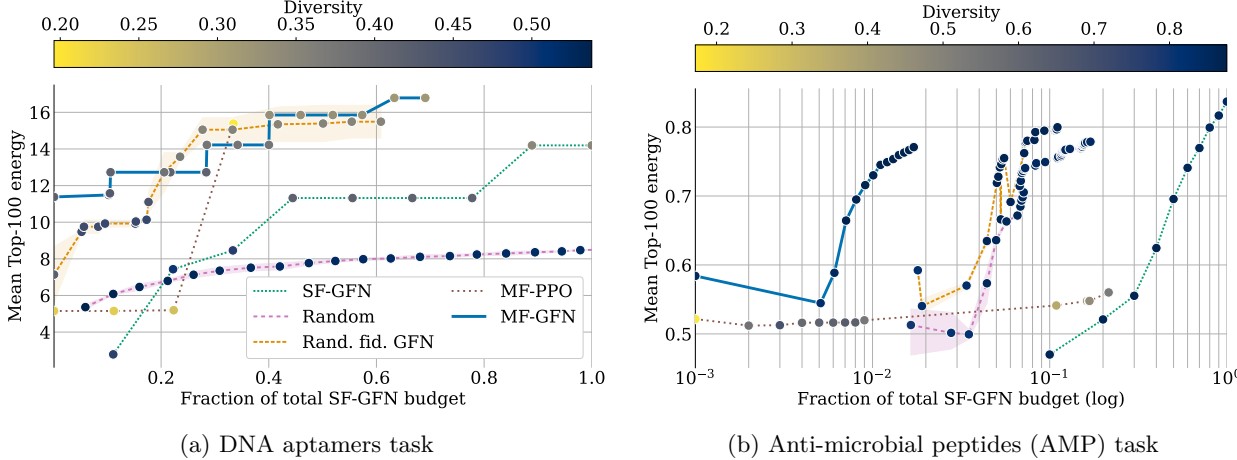

(a) DNA aptamers task        (b) Anti-microbial peptides (AMP) task

Figure 2: Results on the DNA aptamers and AMP tasks. The curves indicate the mean energy $f_M$ within the top-100 samples computed at the end of each active learning round and plotted as a function of the budget used. The colour of the round markers indicates the diversity within the batch (darker colour indicating higher diversity), computed as the average pairwise sequence identity distance (see Appendix D). In both the DNA and the AMP tasks, MF-GFN outperforms all baselines in terms of cost efficiency, while obtaining great diversity in the final batch of top-$K$ candidates.

**GFlowNet with random fidelities (Random fid. GFN)**    Variant of SF-GFN where the candidates are generated with the GFlowNet but the multi-fidelity acquisition function is evaluated with random fidelities sampled from a uniform distribution. This allows us to investigate the contribution of learning to sample the fidelity for each object with GFlowNets.

**Random candidates and fidelities ranked by the acquisition function (Random)**    Quasi-random approach where both candidates and fidelities are randomly sampled from a uniform distribution. We first sample $N$ random $(x, m)$ pairs, then select the top $B$ according to the acquisition function. This baseline is typically competitive due to the value provided by the acquisition.

**Multi-fidelity PPO (MF-PPO)**    Instantiation of multi-fidelity Bayesian optimisation where the acquisition function is optimised using proximal policy optimisation (PPO, Schulman et al., 2017). Unlike with the other baselines, we include an initialisation of $n/3$ steps where $n$ is the maximum number of steps allowed. This is to help exploration and diversity, since without it PPO tends to collapse to generation of very similar candidates.

### 4.3 Benchmark Tasks

As a proof of concept, we perform experiments on two low-dimensional synthetic functions: Branin and Hartmann, widely used in the multi-fidelity Bayesian optimisation literature (Perdikaris et al., 2017; Song et al., 2018; Kandasamy et al., 2019; Li et al., 2020; Folch et al., 2023). These tasks show that MF-GFN is able to obtain results comparable to other multi-fidelity BO methods. We provide these results in Appendix C.4. Nonetheless, the motivation of our work is the challenges posed by large, structured and high-dimensional problems common in scientific discovery. Therefore, in order to assess the performance of MF-GFN on such scenarios, we evaluate it on more complex tasks of practical scientific relevance. We present results on a variety of discovery domains: DNA aptamers (Section 4.3.1), anti-microbial peptides (Section 4.3.2) and small molecules (Section 4.3.3).

### 4.3.1 DNA Aptamers

DNA aptamers are single-stranded nucleotide sequences of nucleobases A, C, T and G, with multiple applications in polymer design due to their specificity and affinity as sensors in crowded biochemical environments (Zhou et al., 2017; Corey et al., 2022; Yesselman et al., 2019; Kilgour et al., 2021). The objective is to maximize the (negative) free energy of the secondary structure of DNA sequences. This free energy can be seen as a proxy of the stability of the sequences. We compute the diversity as one minus the mean pairwise sequence identity among a set of DNA sequences (see Appendix D for the details).

**Setting**   In our experiments, we consider fixed-length sequences of 30 bases and design a GFlowNet environment where the action space $\mathbb{A}$ consists of the choice of base to append to the sequence, starting from an empty sequence. This yields a design space of size $|\mathcal{X}| = 4^{30}$ (ignoring the selection of fidelity in MF-GFN). Further details about the task are discussed in Appendix C.5.1.

**Oracles**   NUPACK (Zadeh et al., 2011), a nucleic acid structure analysis software, is used as the highest fidelity oracle, $f_M$. As a low fidelity oracle, we trained a transformer model on 1 million randomly sampled sequences annotated with $f_M$, and assigned it a cost $100\times$ smaller than the highest-fidelity oracle. The cost difference is selected to simulate practical scenarios where wet lab experiments take hours for evaluation, while cheap online simulations take a few minutes.

**Results**   As presented in Fig. 2a, MF-GFN reaches the best mean top-$K$ energy achieved by its single-fidelity counterpart with just about 25 % of the budget. It is also more efficient than GFlowNet with random fidelities and MF-PPO. Crucially, we also see that MF-GFN maintains a high level of diversity (0.32), even after converging to the top-K scores. On the contrary, MF-PPO (0.20) is not able to discover diverse samples, as is expected based on prior work (Jain et al., 2022).

### 4.3.2 Antimicrobial Peptides

Antimicrobial peptides are short protein sequences which possess antimicrobial properties. As proteins, these are sequences of amino-acids—a vocabulary of size 20 along with a special stop token. The aim is to identify sequences with a high antimicrobial activity, as measured by a model trained on DBAASP (Pirtskhalava et al., 2021). The diversity calculation is equivalent to that of DNA.

**Setting**   We consider variable-length protein sequences with up to 50 residues. Analogous to DNA, if we ignore the fidelity, this yields a design space of size $|\mathcal{X}| > 20^{50}$.

**Oracles**   We construct a three-oracle setup by training deep learning models with different capacities on exclusive subsets of data points. We simulated a setup wherein the two lower fidelity oracles are trained on specific subgroups of the peptides. Similar to the DNA experiment, the lower-fidelity oracles had both a cost $100\times$ less than the highest fidelity oracle. Additional details can be found in Appendix C.5.2.

**Results**   Fig. 2b indicates that in this task MF-GFN obtains even greater advantage over all other baselines in terms of cost-efficiency. It reaches the same maximum mean top-$K$ score as the random baselines with $10\times$ less budget and almost $100\times$ less budget than SF-GFN. In this task, MF-PPO did not achieve comparable results. Crucially, the diversity of the final batch found by MF-GFN stayed high (0.87).

### 4.3.3 Small Molecules

Molecules are clouds of interacting atoms described by a set of quantum mechanical properties. These properties dictate their chemical behaviours and applications. To demonstrate the capability of MF-GFN in the setting of quantum chemistry, we consider two tasks in molecular electronic potentials: maximisation of the (negative) adiabatic ionisation potential (IP) and of the adiabatic electron affinity (EA). These electronic potentials dictate the molecular redox chemistry, and are crucial in organic semiconductors, photoredox catalysis and organometallic synthesis. In this task, the diversity measure is the average pairwise Tanimoto distance among the top-$K$ scoring molecules  (Bajusz et al., 2015), as detailed in Appendix D.

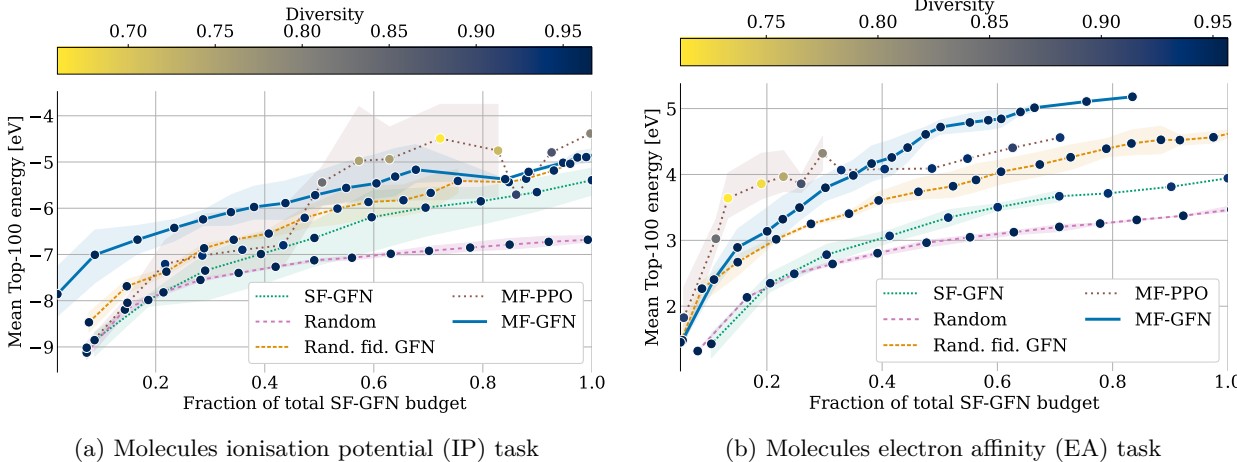

(a) Molecules ionisation potential (IP) task

(b) Molecules electron affinity (EA) task

Figure 3: Results on the molecular discovery tasks: (a) ionisation potential (IP), (b) electron affinity (EA). These visualisations are analogous to those in Fig. 2. The diversity of molecules is computed as the average pairwise Tanimoto distance (see Appendix D). Results generally show MF-GFN's faster convergence in discovering diverse molecules with desirable properties.

**Setting**  We designed the GFlowNet state space by considering variable length sequences of SELFIES tokens (Krenn et al., 2020) to represent molecules, with a vocabulary size of 26. The maximum length was 64, resulting in a design space of $|\mathcal{X}| > 26^{64}$.

**Oracles**  Numerous approximations of these quantum mechanical properties have been developed with different methods at different fidelities, with the famous example of Jacob's ladder in density functional theory (Perdew & Schmidt, 2001). We employed three oracles that correlate with experimental results as approximations of the scoring function, by using various levels of geometry optimisation to obtain approximations to the adiabatic geometries. The calculation of IP or EA was carried out with the semi-empirical quantum chemistry method XTB (Neugebauer et al., 2020). These three oracles had costs of 1, 3 and 7 (respectively), proportional to their computational running demands. See Appendix C.5.3 for further details.

**Results**  The realistic configuration and practical relevance of these tasks allow us to draw stronger conclusions about the usefulness of multi-fidelity active learning with GFlowNets in scientific discovery applications. As in the other tasks evaluated, we here also found MF-GFN to achieve better cost efficiency at finding high-score top-$K$ molecules (Fig. 3), especially for ionisation potentials (Fig. 3a). By clustering the generated molecules, we find that MF-GFN captures as many modes as random generation, far exceeding that of MF-PPO. Indeed, while MF-PPO is able to quickly optimise the target function in the task of electron affinity (Fig. 3b), all generated molecules were from a few clusters (low diversity), which is of much less utility for chemists.

## 4.4   Ablation studies and additional results

Besides the main experiments presented above, we carried out additional experiments to gain further insights about MF-GFN and study the influence of its various components. We provide detailed results in Appendix E and summarise the main conclusions here:

- Analysing the results in terms of the top-$K$ *diverse* samples confirms that the GFlowNet-based approaches are able to jointly optimize scores and diversity, while RL approaches trade diversity for high scores (Appendix E.1).

- As is expected, the advantage of MF-GFN over its single-fidelity counterpart decreases as the cost of the lower fidelity oracles increases. Nonetheless, even with a cost ratio of $1 : 2$ in the DNA task, MF-GFN still outperforms all other methods (Appendix E.2).

- The same conclusions hold for various values of the final candidate set size, $K \in \{50, 100, 200\}$ (Appendix E.3).

- Similar results to the ones presented above were obtained on proof-of-concept experiments with the synthetic functions Branin and Hartmann, common in the multi-fidelity Bayesian optimisation literature (Appendix C.4).

- We expect MF-GFN to query cheap oracles to prune the input space and costly oracles for high-reward candidates. We validate this through a visualisation using the two-dimensional Branin function (Appendix E.5).

- A slightly variation of the baseline Rand. fid. GFN where the fidelities are sampled from a distribution proportional to the cost indicates that learning the fidelity with MF-GFN is still advantageous (Appendix E.6).

## 5 Conclusions, Limitations and Future Work

In this paper, we have presented MF-GFN, a multi-fidelity active learning algorithm that leverages GFlowNets to achieve exploration with diversity for scientific discovery applications. MF-GFN samples candidates as well as the fidelity at which the candidate is to be evaluated, when multiple oracles are available with varying fidelities and costs. We evaluated MF-GFN on benchmark tasks of practical relevance, such as DNA aptamer generation, antimicrobial peptide and small molecule design. Through comparisons with previously proposed methods as well as with variants of our method designed to understand the contributions of different components, we conclude that multi-fidelity active learning with GFlowNets not only outperforms its single-fidelity active learning counterpart in terms of cost effectiveness and diversity of sampled candidates, but it also offers an advantage over other multi-fidelity methods due to its ability to learn a stochastic policy to jointly sample objects and the fidelity of the oracle.

**Limitations and Future Work** Aside from the molecular modelling tasks, our empirical evaluations in this paper involved simulated oracles with manually selected costs. Future work should evaluate MF-GFN with more practical oracles and costs that reflect their computational or financial demands. As a first approach to multi-fidelity active learning with GFlowNets, the proposed algorithm may be improved and adapted to specific applications in different ways. For example, instead of selecting the top-$B$ candidates at the end of each active learning round, one could adopt stochastic batch acquisition strategies (Kirsch et al., 2023). If the oracle costs reflect run-time, as is the case in many scientific discovery problems, then it would be interesting to consider strategies that would not block the exploration while waiting for the most expensive oracle. Furthermore, a promising avenue that we do not study in this paper is the application of MF-GFN in more complex, structured design spaces, such as hybrid (discrete and continuous) domains (Lahlou et al., 2023; AI4Science et al., 2023), as well as both multi-fidelity and multi-objective problems (Jain et al., 2023b).

Finally, given the complexity of multi-fidelity active learning algorithms, consisting of multiple components, we have barely optimised the many hyperparameters and design choices. In fact, we have opted for conservative, simple choices in several cases. For example, as oracle confidence we have simply used the inverse of the cost instead of a more accurate estimate of the oracle fidelity. Therefore, we expect significant room for improvement from simple hyperparameter optimisation.

## Statement of Broader Impact

Our work is motivated by pressing challenges to sustainability and public health, and we envision applications of our approach to drug discovery and materials discovery. However, as with all work on these topics, there

is a potential risk of dual use of the technology by nefarious actors (Urbina et al., 2022). The authors strongly oppose any uses or derivations of this work intended to cause harm to humans or the environment and explicitly request the careful consideration of potential harms.

## Reproducibility Statement

We have made an effort to include the most relevant details of our proposed algorithm in the main body of the paper. For example, a detailed procedure of the steps of the algorithm is presented in Algorithm 1. Besides this, we have included additional details about the algorithm in Appendices A and B. We have also provided the most relevant information about the experiments in Section 4, for instance including a description of the data representation and the oracles for each of the benchmark tasks. The rest of the details about the experiments are provided in Appendix C for the sake of better clarity, transparency and reproducibility. Finally, the implementation of MF-GFN and the code to reproduce the experiments is publicly available in a GitHub repository:



`https://github.com/nikita-0209/mf-al-gfn`

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

## A   MF-GFN Algorithm

Our algorithm, MF-GFN, detailed in Algorithm 1, proceeds as follows: An active learning round $j$ starts with a data set of annotated samples $\mathcal{D}_j = \{(x_i, f_{m_i}(x_i), m_i)\}_{1 \leq m \leq M}$. If no initial data set is available, a set of data points could be sampled randomly. The data set is used to fit a probabilistic multi-fidelity surrogate model $h(x, m)$ of the posterior $p(f_m(x)|x, m, \mathcal{D})$. The output of the surrogate model is then used to compute the value of a multi-fidelity acquisition function $\alpha(x, m)$. In our experiments, we use the multi-fidelity version (Takeno et al., 2020) of max-value entropy search (MES, Wang & Jegelka, 2017), which is an information-theoretic acquisition function widely used in Bayesian optimisation. Additional details about the surrogate and the acquisition function are provided Section 3 and Appendix B.

The acquisition function is used to derive the reward to train a multi-fidelity GFlowNet (see details in Appendix C.3. The GFlowNet is trained to sample tuples $(x, m)$ proportionally to the reward. An active learning round terminates by generating $N$ tuples from the sampler (here the GFlowNet policy $\pi$) and forming a batch with the best $B$ objects, according to $\alpha$. Note that $N \gg B$, since sampling from a GFlowNet is relatively inexpensive. The selected objects are annotated by the corresponding oracles and incorporated into the data set, such that $\mathcal{D}_{j+1} = \mathcal{D}_j \cup \{(x_1, f_{m_1}(x_1), m_1), \ldots, (x_B, f_{m_B}(x_B), m_B)\}$. This procedure is repeated, progressively acquiring new data, until the budget is exhausted.

## B   Surrogate Models and Acquisition Function

In this section, we discuss additional details about the surrogate model and acquisition function used in our experiments.

### B.1   Gaussian Processes

Following the Bayesian optimisation literature, we assume a Gaussian Process (GP) prior on the joint distribution over the function values at different fidelities. The posterior distribution over the oracles $p(f_m(x)|x, m, \mathcal{D})$ is a also a GP. Concretely, consider a set of n points $\{(x_1, m_1), (x_2, m_2), \ldots, (x_n, m_n)\}$ with observed values $\{f_{m_1}(x_1), \ldots f_{m_n}(x_n)\}$. Let $\mathcal{D} = \{(x_1, f_{m_1}(x_1), m_1), \ldots, (x_n, f_{m_n}(x_n), m_n)\}$. A key decision for modelling with GPs is the choice of the kernel. As discussed in Section 3.2, we use the linear truncated multi-fidelity kernel previously used by Mikkola et al. (2023). Assuming a GP prior with mean function $\mu$, kernel $K_{MF}$ (Eq. (2)) and an additive Gaussian noise observation model with variance $\sigma$, the posterior distribution conditioned on $\mathcal{D}$ is also a GP with the following mean and covariance:

$$\begin{aligned}\mu_n(x, m) = {} & \mu(x_{1:n}, m_{1:n}) + K_{MF}((x, m), (x_{1:n}, m_{1:n}))(K_{MF}((x_{1:n}, m_{1:n}), (x_{1:n}, m_{1:n})) \\ & + \sigma^2 I)^{-1}(f_m(x_{1:n}) - \mu(x_{1:n}, m_{1:n})).\end{aligned}$$

$$\begin{aligned}K_n((x_i, m_i), (x_j, m_j)) = {} & K_{MF}((x_i, m_i), (x_j, m_j)) \\ & - K_{MF}((x_i, m_i), (x_{1:n}, m_{1:n}))(K_{MF}((x_{1:n}, m_{1:n}), (x_{1:n}, m_{1:n})) \\ & + \sigma^2 I)^{-1} K_{MF}((x_{1:n}, m_{1:n}), (x_j, m_j)).\end{aligned}$$

### B.2   Deep Kernel Learning

For synthetic tasks, we use an exact GP for the surrogate. However, for larger, higher dimensional tasks, having a kernel which captures the structure in the space becomes critical as standard kernels such as RBF and Matérn have difficulty to accurately capture a useful notion of similarity. Thus, for the larger benchmark tasks we use deep kernel learning (DKL; Wilson et al., 2016) to alleviate the challenges of defining a kernel. DKL involves using a deep neural network $g_\omega$, where $\omega$ are the parameters, to learn a low-dimensional embedding $z$ for an input $x$, $z = g_\omega(x)$ and applying the kernel on these low-dimensional embeddings.

$$K_X(x_i, x_j) \rightarrow K_X(g_\omega(x_i), g_\omega(x_j)),$$

where $g_\omega$ is a transformer for our experiments. Additionally, to scale the GP to large datasets, we implement the stochastic variational GP based on the greedy inducing point method (Chen et al., 2018). Training surrogate models with DKL can lead to some instabilities, so for our experiments we rely on recommendations from Stanton et al. (2022) to ensure stable training of the surrogate model.

### B.3 Acquisition Function

In our experiments, we use the max-value entropy search (MES, Wang & Jegelka, 2017) acquisition function. Specifically, we use the multi-fidelity MF-MES variant proposed by Takeno et al. (2020), which for a given candidate $(x, m)$ measures the mutual information between the value of $f_m(x)$ and the maximum value of the highest fidelity oracle $f_M^*$.

$$\alpha(x, m) = I(f_M^\star; f_m(x)|\mathcal{D}_j) = H(f_m(x)|\mathcal{D}_j) - \mathbb{E}_{f^\star}[H(f_m(x)|f_M^\star, \mathcal{D}_j)|\mathcal{D}_j].$$

The acquisition function can also be interpreted as the expected information gain about $f_M^*$ obtained by querying $f_m(x)$, where information gain is defined as the reduction in entropy over $f_M^*$ induced by observing $f_m(x)$.

The mutual information can be expensive to compute and is typically approximated using Monte Carlo samples (Wang & Jegelka, 2017). For our experiments, we use the GIBBON approximation (Moss et al., 2021). Moss et al. (2021) proposed an approximation to the MES acquisition function that is tractable and efficient to compute. Concretely, it takes the following form:

$$\alpha(x, m) \approx \frac{1}{\lambda_m} \frac{1}{|\mathcal{F}|} \sum_{F \in \mathcal{F}} \text{IG}^{\text{Approx}}(f_m(x), F|\mathcal{D}_n), \text{with}$$

$$\text{IG}^{\text{Approx}} = \frac{1}{2} \log |R| - \frac{1}{2|\mathcal{F}|} \sum_{F \in \mathcal{F}} \log \left(1 - \rho^2 \frac{\phi(\gamma(F))}{\Phi(\gamma(F))} \left[\gamma(F) + \frac{\phi(\gamma(F))}{\Phi(\gamma(F))}\right]\right),$$

where $\phi$ and $\Phi$ are the standard normal cumulative distribution and probability density functions (arising from the expression for the differential entropy of a truncated Gaussian), $\gamma(F) = \frac{F - \mu_n(x, m)}{\sigma_n(x, m)}$, $R$ is the correlation matrix with elements $R_{i,j} = \frac{\Sigma_{i,j}}{\Sigma_{i,i}\Sigma_{j,j}}$, $\rho = \text{Corr}(f_m(x), f_M(x))$, and $\mathcal{F} = \{F_i\}_{i=1}^{|\mathcal{F}|}$, $F \sim p(f_M^*|\mathcal{D})$. $\mu_n(x, m)$ is the predictive mean, $\sigma_n(x, m)$ is the predictive standard deviation and $\Sigma_{i,j}$ is the predictive covariance of the surrogate model for fidelities $i, j$ at $x$.

Note that our algorithm is not bound to a particular acquisition function (or surrogate model) and therefore these components may be adapted to the specific needs of the application.

## C Experimental Details

This section presents the details about the experiments discussed in Section 4. First, we provide general details about all tasks and then present details specific to each task in separate sections.

### C.1 Initial data set and budget

We define a budget ($\Lambda_0$) for the initial data set. Let $\lambda_m$ be the cost of evaluating $x$ with oracle $f_m$, and $n_{SF}$, $n_{MF}$ be the number of initial training points in the single- and multi-fidelity experiments respectively. Also let $n_m$ be the number of training points evaluated against $f_m$ in the multi-fidelity experiment such that $n_{MF} = \sum_{m=1}^{M} n_m$. Then,

$$\Lambda_0 = n_{SF} \times \lambda_M = \sum_{m=1}^{m=M} n_m \times \lambda_m.$$

The initial data set is split into train and validation in the ratio of 9:1 for all tasks. Task-specific information is summarized in Table 1.

For each task, we assign a total active learning budget $\Lambda = \gamma \times \lambda_M$ (Table 2). $\gamma$ was selected based on the rate of convergence of the algorithms to the modes. Note that during an active learning round, only the oracle evaluations of the sampled batch contribute to $\Lambda$. The cost of sampling from a trained GFlowNet is nearly negligible compared to the oracle evaluations. This is why we can afford to sample a large number of samples ($N = 5 \times B$) to then select the best $B$, according to the acquisition function (Algorithm 1).

## C.2  DKL Implementation Details

Here, we describe our implementation of DKL, which is inspired by Stanton et al. (2022).

**Neural Network Architecture**  For all experiments, the same base architecture was used, featuring transformer encoder layers with position masking for padding tokens. Standard pre-activation residual blocks were included, comprising two convolutional layers, layer normalisation, and swish activations. The encoder embeds input sequences with standard vocabulary and sinusoidal position embeddings. The encoder is trained with the Masked Language Modeling (MLM) objective which is calculated by randomly masking input tokens and subsequently computing the empirical cross-entropy between the original sequence and the predictive distribution generated by the MLM head for the masked positions.

**Optimiser Hyperparameters**  The running estimates of the first two moments in the Adam optimiser (Kingma & Ba, 2015) were disabled by setting $\beta_1 = 0.0$ and $\beta_2 = 0.01$.

**Kernel Hyperparameters**  In order to force the encoder to learn features appropriate for the initial lengthscale, we place a tight Gaussian prior with $\sigma = 0.01$ around the intial lengthscale value. The reinitialisation procedure for inducing point locations and variational parameters outlined by Maddox et al. (2021) was followed.

## C.3  Reward Function and Policy Models Details

**Neural Network Architecture**  For all tasks, the architecture of the GFlowNet forward policy ($P_F$) model is a multi-layer perceptron with 2 hidden layers and 2048 units per layer. The backward policy ($P_B$) model was set to share all but the last layer parameters with $P_F$. We use LeakyReLU as our activation function as in Bengio et al. (2021a). All models are trained with the Adam optimiser (Kingma & Ba, 2015).

**Reward Function**  As detailed in Algorithm 1, the GFlowNet is trained to generate samples with a higher value of the MES acquisition function and its multi-fidelity variant in single- and multi-fidelity experiments, respectively. In order to increase the relative reward of higher values of the acquisition function, we scale the MES value $\alpha(x, m)$ by $\frac{1}{\beta}$, with $0 < \beta \leq 1$. On an additional note, MES exhibits increased sparsity as more samples are discovered. Hence, in order to facilitate optimisation, we introduce another scaling factor denoted by $\rho^{j-1}$, which depends on the active learning round $j$. Altogether, our GFlowNet reward function is the following:

$$R(\alpha(x, m), \beta, \rho, j) = \frac{\alpha(x, m) \times \rho^{j-1}}{\beta},$$

Note that within an active learning round, the GFlowNet samples (approximately) from this fixed reward function and thus the policy need not be conditioned on $j$. The hyperparameter values for all tasks are detailed in Table 2.

Our models are implemented in PyTorch (Paszke et al., 2019), and rely on BoTorch (Balandat et al., 2020) and GPyTorch (Gardner et al., 2021).

## C.4  Synthetic Tasks

This section describes the experiments and results obtained on the synthetic tasks Branin and Hartmann, included for completeness, to allow a comparison with the traditional Bayesian optimisation literature.

Table 1: Oracle costs (indexed by increasing level of fidelity) and initial data set details.

| Task | Oracle Costs | | | Initial Data Set | | | | |
| | $\lambda_1$ | $\lambda_2$ | $\lambda_M$ | $\Lambda_0$ | $n_{SF}$ | $n_{MF}$ | | |
| | | | | | | $n_1$ | $n_2$ | $n_M$ |
|---|---|---|---|---|---|---|---|---|
| Branin | 0.01 | 0.1 | 1 | 4 | 4 | 20 | 20 | 2 |
| Hartmann 6D | 0.125 | 0.25 | 1 | 25 | 25 | 80 | 40 | 5 |
| DNA | – | 0.2 | 20 | 1600 | 80 | – | 3000 | 50 |
| AMP | 0.5 | 0.5 | 50 | 2500 | 50 | 2000 | 2000 | 10 |
| Molecules | 1 | 3 | 7 | 1050 | 150 | 700 | 68 | 16 |

Table 2: Hyperparameters concerning the active learning setting and the policy reward function.

| Task | Surrogate Model | Active-learning | | Policy reward function | |
| | | $\gamma$ | $B$ | $\beta$ | $\rho$ |
|---|---|---|---|---|---|
| Branin | Exact GP | 300 | 30 | 1 | 1 |
| Hartmann 6D | Exact GP | 100 | 10 | 1e-2 | 1 |
| DNA | DKL | 256 | 512 | 1e-5 | 2 |
| Antimicrobial Peptides | DKL | 20 | 32 | 1e-5 | 1 |
| Molecules | DKL | 180 | 128 | 1e-6 | 1.5 |

### C.4.1 Branin

We consider an active learning problem in a two-dimensional space $(x_1, x_2)$ where the target function $f_M$ is the Branin function, as modified by Sobester et al. (2008) and implemented in BoTorch (Balandat et al., 2020). In the standard domain $[-5, 10] \times [0, 15]$, the Branin function has three modes and is evaluated using the following expression:

$$f(x) = (x_2 - \frac{-1.25x_1^2}{\pi^2} + \frac{5x_1}{\pi} - 6)^2 + (10 - \frac{5}{4\pi})\cos(x_1) + 10.$$

This corresponds to the modification introduced by Sobester et al. (2008). As lower fidelity functions, we used the expressions from Perdikaris et al. (2017), which involve non-linear transformations of the true function as well as shifts and non-uniform scalings. The functions, indexed by increasing level of fidelity, are the following:

$$f_1(x) = f_2(1.2(x + 2)) - 3x_2 + 1,$$

$$f_2(x) = 10\sqrt{f(x - 2)} + 2(x_1 - 0.5) - 3(3x_2 - 1) - 1.$$

This amounts to three levels of fidelity, including the true function. The lower-fidelity oracles, the costs of the oracles $(0.01, 0.1, 1.0)$ as well as the number of points queried in the initial training set were adopted from Li et al. (2020).

In order to consider a discrete design space, we interpolate the domain into a discrete $100 \times 100$ grid. We model this grid with a GFlowNet as in (Bengio et al., 2021a; Malkin et al., 2022): starting from the origin $(0, 0)$, for any state $s = (x_1, x_2)$, the action space consists of the choice between the exit action or the dimension to increment by 1, provided the next state is in the limits of the grid.

We use the BoTorch implementation of an exact multi-fidelity Gaussian process as described in Appendix B.1 for regression. The active learning acquisition batch size $B$ is 30 in the Branin task.

Fig. 4a illustrates the results for this task. We observe that MF-GFN is able to reach the minimum of the Branin function with a smaller budget than the single-fidelity counterpart and the baselines.

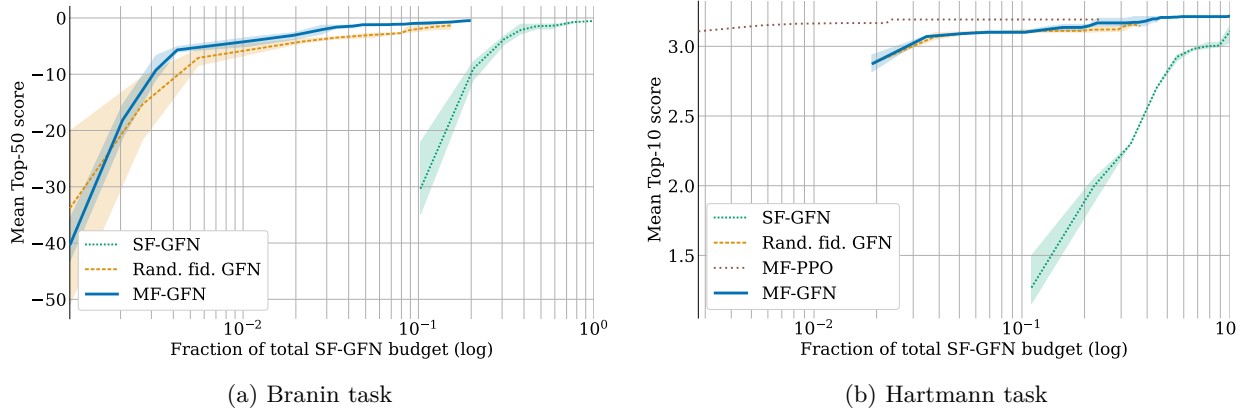

|                    |                    |
| :----------------: | :----------------: |
| (a) Branin task    | (b) Hartmann task  |

Figure 4: Results on the synthetic tasks—Branin and Hartmann functions. The curves indicate the mean score $f_M$ within the top-50 and top-10 samples (for Branin and Hartmann, respectively) computed at the end of each active learning round and plotted as a function of the budget used. The random baseline is omitted from this plot to facilitate the visualisation since the results were significantly worse in these tasks. We observe that MF-GFN clearly outperforms the single-fidelity counterpart (SF-GFN) and slightly improves upon the GFlowNet baseline that samples random fidelities. On Hartmann, MF-PPO initially outperforms the other methods.

### C.4.2  Hartmann 6D

We consider the 6-dimensional Hartmann function as objective $f_M$ on a hyper-grid domain. It is typically evaluated on the hyper-cube $x_i \in [0,1]^6$ and consists of six local maxima. The expression of the true Hartmann function is given by

$$f(x) = \sum_{i=1}^{4} \eta_i exp(-\sum_{j=1}^{3} A_{ij}(x_j - P_{ij})^2),$$

where $\eta = [1.0, 1.2, 3.0, 3.2]$ and $A, P \in \mathbb{R}^{4 \times 6}$ are the following fixed matrices:

$$A = \begin{bmatrix} 10 & 3 & 17 & 3.5 & 1.7 & 8 \\ 0.05 & 10 & 17 & 0.1 & 8 & 1 \\ 3 & 3.5 & 1.7 & 10 & 17 & 8 \\ 17 & 8 & 0.05 & 10 & 0.1 & 1 \end{bmatrix},$$

$$P = 10^{-4} \times \begin{bmatrix} 3689 & 1170 & 267 \\ 4699 & 4387 & 7470 \\ 1091 & 8732 & 5547 \\ 381 & 5743 & 8828 \end{bmatrix}.$$

To simulate the lower fidelities, we modify $\eta$ to $\eta(m)$ where $\eta(m) = \eta + \delta(M - m)$ where $\delta = [0.01, -0.01, -0.1, 0.1]$ and $M = 3$. The domain is $X = [0, 1]^6$. This implementation was adopted from Kandasamy et al. (2019). As with Branin, we consider three oracles, adopting the lower-fidelity oracles and the set of costs $(0.125, 0.25, 1.0)$ from Song et al. (2018).

We discretise the domain into a six-dimensional hyper-grid of length 10, yielding $10^6$ possible candidate points. For the surrogate, we use the same exact multi-fidelity GP implementation as of Branin. The active learning acquisition batch size $B$ is 10.

The results for the task are illustrated in Fig. 4b, which indicate that multi-fidelity active learning with GFlowNets (MF-GFN) offers an advantage over single-fidelity active learning (SF-GFN) as well as some of

the other baselines in this higher-dimensional synthetic problem as well. The better performance on MF-PPO can be attributed to the fact that while the GFN initiates its exploration from the origin point, the PPO commences from a random starting point within a bounded range, allowing at most three units of displacement (maximum possible displacement is 10 units) along each of the six axes. We hypothesise that this aids the PPO algorithm in expediting the discovery of modes within the optimisation process. While MF-PPO performs better in this task, as shown in the benchmark experiments, it tends to collapse to single modes of the function in complex high-dimensional scenarios.

## C.5  Benchmark Tasks

### C.5.1  DNA

We conduct experiments using a two-oracle setup $(f_M, f_1)$ with costs $\lambda_M = 20$ and $\lambda_1 = 0.2$ for the high and low fidelity oracles, respectively. As $f_M$, we use the free energy of the secondary structure of DNA sequences obtained via the software NUPACK (Zadeh et al., 2011), setting the temperature at 310 K. $f_1$ is a transformer (with 8 encoder layers, 1024 hidden units per layer and 16 heads) trained on 1 million random sequences annotated by $f_M$. To evaluate the performance of $f_1$ (with respect to $f_M$), we construct a test set by sampling sequences from a uniform distribution of the free energy. On this test set, the explained variance of the $f_1$ is calculated to be 0.8. For the probabilistic surrogate model, we implement deep kernel learning, the hyperparameters of which are provided in Table 4. The active learning acquisition batch size $B$ is 512.

### C.5.2  Antimicrobial Peptides

We use data from DBAASP (Pirtskhalava et al., 2021), containing antimicrobial activity labels, which is originally split into three sets: $D_1$ for training the oracle, $D_2$ as the initial data set in the active learning loop and $D_3$ as the test set (Jain et al., 2022).

Table 3: Oracles for the antimicrobial peptides task.

| Oracle | Training points | Model | Layers | Hidden units | Training Epochs |
|--------|-----------------|-------|--------|--------------|-----------------|
| $f_1$ | 3447 | MLP | 2 | 512 | 51 |
| $f_2$ | 3348 | MLP | 2 | 512 | 51 |
| $f_M$ | 6795 | MLP | 2 | 1024 | 101 |

We design a three-oracle setup $(f_M, f_2, f_1)$ where each oracle is a different neural network model. The configurations of the oracle models are presented in Table 3. Biologically, each antimicrobial peptide can be classified into an antimicrobial group. $f_M$ is trained on the entire dataset $D_1$. However, for $f_1$ and $f_2$, we divide $D_1$ into two approximately equally-sized disjoint subsets. This simulated a setup wherein each lower fidelity oracle specialised in different sub-regions of the entire sample space. We set costs $\lambda_M = 50$ and $\lambda_1 = \lambda_2 = 0.5$ as $f_1$ and $f_2$ have similar configurations. The explained variance of $f_1$ and $f_2$ (with respect to $f_M$) on a uniform test set, $D_3$ was 0.1435 and 0.099 respectively. For the surrogate, we implement deep kernel learning with the hyperparameters detailed in Table 4. The active learning acquisition batch size $B$ is 32.

Table 4: Deep kernel hyperparameters for the DNA and antimicrobial tasks.

| Architecture hyperparameter | Value | Optimisation hyperparameter | Value |
|-----------------------------|-------|-----------------------------|-------|
| Number of layers | 8 | Batch size | 128 |
| Number of heads | 8 | Learning rate | 1e-3 |
| Latent dimension | 64 | Adam EMA parameters $(\beta_1, \beta_2)$ | (0.0, 1e-2) |
| GP likelihood variance init. | 0.25 | Maximum number of epochs | 512 |
| GP length scale prior | $\mathcal{N}(0.7, 0.01)$ | Early stopping patience | 15 |
| Number of SGVP inducing points | 64 | Early stopping holdout ratio | 0.1 |

### C.5.3 Small Molecules

For the experiments with small molecules, we construct a three oracle setup ($f_M$, $f_2$, $f_1$) with costs representing the actual compute time. We implement the oracles using RDKit 2023.03 (rdk, 2023) and the semi-empirical quantum chemistry package `xTB`. We use GFN2-xTB (Bannwarth et al., 2019) method for the single point calculation of ionisation potential (IP) and electron affinity (EA) with empirical correction terms.

In $f_1$, we consider one conformer obtained by RDKit with its geometry optimised via the force-field MMFF94 (Halgren, 1996). This geometry is used to calculate (vertical) IP/EA. In $f_2$, we consider two conformers obtained by RDKit, and take the lowest energy conformer after optimisation by MMFF94, and further optimise it via GFN2-xTB to obtain the ground state geometry; this remains a vertical IP/EA calculation. In $f_M$, we consider four conformers obtained by RDKit, and take the lowest energy conformer after optimisation by MMFF94, and further optimise it via GFN2-xTB; the corresponding ion is then optimised by GFN2-xTB, and the adiabatic energy difference is obtained via total electronic energy. The fidelities are based on the fact that vertical IP/EA approximates that of adiabatic ones (to varying degrees, depending on the molecule). On a uniform test set of 1400 molecules, the explained variance of $f_1$ and $f_2$ (with respect to $f_M$) is $0.1359, 0.279$ and $0.79, 0.86$ for the EA and IP tasks respectively.

The surrogate model is a deep kernel with the hyperparameters are provided in Table 5. The active learning acquisition batch size $B$ is 128. In the environment for GFN, we consider a set of SELFIES vocabularies containing aliphatic and aromatic carbon, boron, nitrogen, oxygen, fluorine, sulfur, phosphorous, chlorine, and bromine, subject to standard valency rules.

Table 5: Deep Kernel hyperparameters for the molecular tasks

| Architecture hyperparameter | Value | Optimisation hyperparameter | Value |
| --- | --- | --- | --- |
| Number of layers | 8 | Batch size | 128 |
| Number of heads | 8 | Learning rate | 1e-3 |
| Latent dimension | 32 | Adam EMA parameters ($\beta_1, \beta_2$) | (0.0, 1e-2) |
| GP likelihood variance init. | 0.25 | Maximum number of epochs | 512 |
| GP length scale prior | $\mathcal{N}(0.7, 0.01)$ | Early stopping patience | 15 |
| Number of SGVP inducing points | 64 | Early stopping holdout ratio | 0.1 |

We note that this is proof-of-concept and hence neither do we conduct a full search of conformers, nor do we perform Density Functional Theory calculations. Nonetheless, we observe that the highest fidelity oracle has a good correlation with experiments (Neugebauer et al., 2020). We do not consider synthesisability in this study and we note it may negatively impact GFN as unphysical molecules could produce false results for the semi-empirical oracle.

## D   Metrics

In this section, we provide additional details about the metrics used for the evaluation of the proposed MF-GFN as well as the baselines.

Since we are interested in discovering multiple objects with high scores, instead of only one, in order to compute our evaluation metrics, we consider the set of top-$K$ candidates in the data set $\mathcal{D}$. Formally, let $\mathcal{D}_j{}^1$ be the data set after active learning round $j$:

$$\mathcal{D}_j = \{x_1, x_2, \dots, x_n\},$$

such that $f_M(x_i) \geq f_M(x_j), \forall\ i < j$. Then, we define the set of top-$K$ candidates as

---

[1]Rigorously, the data set consists of triplets $(x_i, f_{m_i}(x_i), m_i)$, as defined in Section 3. Here, for better readability, we simplify the notation and define the data set in terms of the candidates $x_i$ only.

$$\text{top-}K(\mathcal{D}_j) = \{x_i | x_i \in \mathcal{D}_j, f_M(x_i) \geq f_M(x_K), 1 \leq i \leq K\}.$$

Note that, for evaluation purposes, the top-$K$ set is selected according to the highest-fidelity oracle $f_M(x_i)$ and not the oracle selected by the algorithm for the candidate, $f_{m_i}(x_i)$.

**Mean top-$K$ score**   We adapt this metric from Bengio et al. (2021a), which is meant to reflect the ability of the algorithm to discover multiple high-scoring samples. This metric is the mean value of the highest-fidelity oracle across the top-$K$ set:

$$\text{mean(top-}K(\mathcal{D}_j)) = \frac{1}{K} \sum_{i=1}^{K} f_M(x_i). \tag{4}$$

**Top-$K$ diversity**   This metric is meant to reflect the ability of the algorithm to discover diverse samples. In order to measure the diversity of the top-$K$ set of candidates for each of the tasks, we use use one minus the similarity index, which is equal to one if all sequences/molecules in a batch are identical, and zero if all sequences/molecules are maximally different. Specifically,

$$\text{diversity(top-}K(\mathcal{D}_j)) = 1 - \frac{1}{\binom{K}{2}} \sum_{\substack{x_i, x_k \in \mathcal{D}_j \\ x_i \neq x_k}} s(x_i, x_k), \tag{5}$$

where $s(x_i, x_j)$ is a similarity measure between elements $x_i$, and $x_k$.

- **DNA aptamers**: The similarity measure is calculated by the mean pairwise sequence identity within a set of DNA sequences. The higher the sequence identity, the more similar the sequences are. We utilize global alignment with the Needleman-Wunsch algorithm and standard nucleotide substitution matrix, as calculated by the Biotite package (Kunzmann & Hamacher, 2018). Here, $s(x_i, x_k) = \frac{\text{Number of matching nucleotides}}{\text{Total number of nucleotides in the alignment}}$.

- **Antimicrobial peptides**: The similarity measure is calculated by the mean pairwise sequence identity within a set of peptide sequences. The higher the sequence identity, the more similar the sequences are. We utilize global alignment with the Needleman-Wunsch algorithm and BLOSUM62 substitution matrix, as calculated by the Biotite package (Kunzmann & Hamacher, 2018). Here, $s(x_i, x_k) = \frac{\text{Number of matching aminoacids}}{\text{Total number of aminoacids in the alignment}}$.

- **Molecules**: The similarity measure is calculated by the mean pairwise Tanimoto similarity within a set of molecules. The higher the Tanimoto similarity, the more similar the molecules are. Tanimoto metrics are calculated from Morgan Fingerprints (radius of two, size of 2048 bits) as implemented in the RDKit package (rdk, 2023). Here, $s(x_i, x_k) = \frac{|F(x_i) \cap F(x_k)|}{|F(x_i) \cup F(x_k)|}$.

**Mean diverse top-$K$ score**   This is an alternative version to the mean top-$K$ score metric by which we restrict the selection of the $K$ candidates to data points that are diverse between each other. We use similarity measures (*vide infra*) such that we sample the top-$K$ candidates where each candidate is at most similar to each other by a certain threshold. For antimicrobial peptides, the sequence identity threshold is 0.35; for DNA aptamers, the sequence identity threshold is 0.60; for molecules, the Tanimoto similarity distance threshold is 0.35. The results using this metric are reported in Appendix E.

## E   Additional Results

This section includes additional results and ablation studies to complement the analysis presented in the main body of the paper.

### E.1  Energy of Diverse Top-$K$

In this section, we complement the results presented in Section 4 with the mean diverse top-$K$ scores, as defined in Appendix D. This metric combines that mean top-$K$ score and the measure of diversity. Figure 5 shows the results on the DNA, AMP and the molecular tasks.

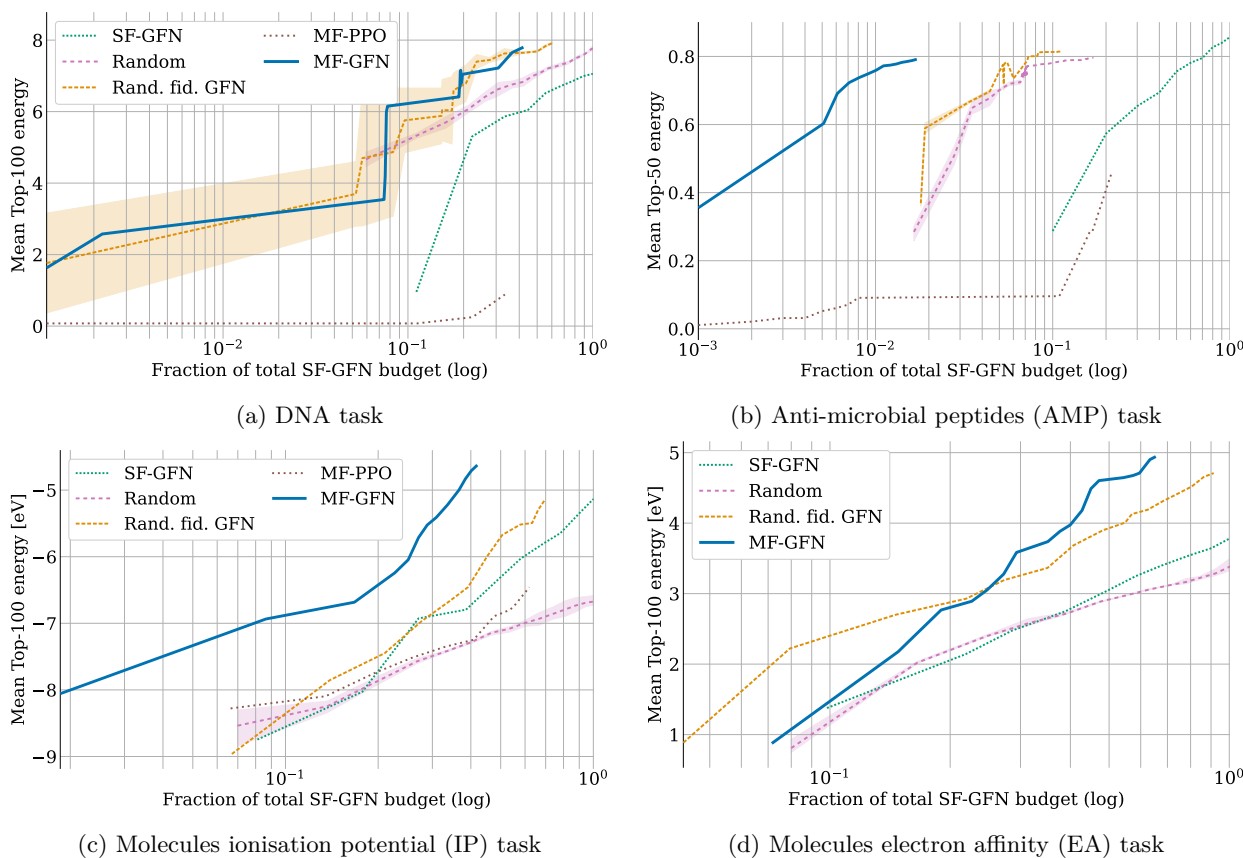

(a) DNA task

(b) Anti-microbial peptides (AMP) task

(c) Molecules ionisation potential (IP) task

(d) Molecules electron affinity (EA) task

Figure 5: Mean scores (energy) of diverse top-$K$ candidates on the DNA (top left), AMP (top right) and molecular (bottom) tasks. The mean energy is computed across the top-$K$ candidates at each active learning round that also satisfy the criteria of diversity. Consistent with the diversity metrics observed in Fig. 2, we here see that GFlowNet-based methods, and especially MF-GFN, obtain good results according to this metric, while MF-PPO achieves comparatively much lower mean energy.

The results with this metric allow us to further confirm that multi-fidelity active learning with GFlowNets is able to discover sets of diverse candidates with high mean scores, as is sought in many scientific discovery applications. In contrast, methods that do not encourage diversity such as RL-based algorithms (MF-PPO) obtain comparatively much lower results with this metric.

### E.2  Impact of Oracle Costs

As discussed in Appendix B.3, a multi-fidelity acquisition function like the one we use—defined in Eq. (3)—is a cost-adjusted utility function. Consequently, the cost of each oracle plays a crucial role in the utility of acquiring each candidate. In our tasks with small molecules (Section 4.3.3), for instance, we used oracles with costs proportional to their computational demands and observed that multi-fidelity active learning largely outperforms single-fidelity active learning. However, depending on the costs of the oracles, the advantage of multi-fidelity methods can significantly diminish.

In order to analyse the impact of the oracle costs on the performance of MF-GFN, we run several experiments on the DNA task (Section 4.3.1), which consists of two oracles, with additional sets of oracle costs. In particular, besides the costs used in the experiments presented in Section 4.3.1, $(0.2, 20)$ for the lowest and highest fidelity oracles, we run experiments with costs $(1, 20)$ and $(10, 20)$. Additionally, we perform similar experiments on the Hartmann task with 5 sets of oracle costs.

The results on the DNA task, presented in Fig. 6a, indeed confirm that the advantage of MF-GFN over SF-GFN decreases as the cost of the lowest-fidelity oracle becomes closer to the cost of the highest-fidelity oracle. However, it is remarkable that even with a ratio of costs as small as $1 : 2$, MF-GFN still outperforms not only SF-GFN but also MF-PPO in terms of cost effectiveness, without diversity being negatively impacted. It is important to note that in practical scenarios of scientific discovery, the cost of lower fidelity oracles is typically orders of magnitude smaller than the cost of the most accurate oracles, since the latter may correspond to wet-lab experiments or expensive computer simulations. The results on the Hartmann task (Fig. 6b) further confirm the above conclusions.

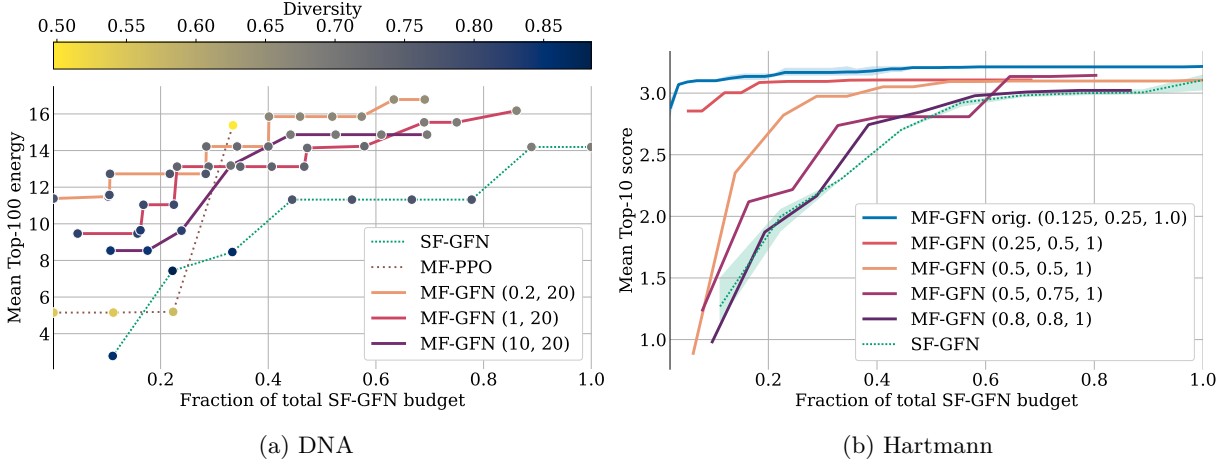

(a) DNA                                     (b) Hartmann

Figure 6: Analysis of the impact of the oracle costs on the performance of MF-GFN on the DNA task and the synthetic Hartmann task. On the DNA task, we observe that the advantage over SF-GFN and MF-PPO $(0.2, 20)$ decreases as the cost of the lower fidelity oracle becomes closer to the cost of the highest fidelity oracle. Nonetheless, even with a cost ratio of $1 : 2$ MF-GFN displays remarkable performance with respect to other methods. Similar conclusions can be drawn from the analysis on the Hartmann task.

## E.3 Impact of the Acquisition Batch Size

We evaluate the impact of the active learning acquisition batch size $B$ on the performance of MF-GFN and its comparison with the baselines for the small molecules IP task with different acquisition batch sizes. From the results in Fig. 7, we notice that the reward curve becomes slightly steeper with larger batch sizes.

## E.4 Impact of the Choice of the Final Candidate Set Size

For the set of results presented in the main paper, we computed the mean top-$K$ energy and diversity on the final batch of candidates of size $K = 100$. While the choice of $K$ is not arbitrary as it is related to the active learning acquisition size and in turn to reasonable numbers in the domains of application, it is interesting to study whether our conclusions are robust to other choices of $K$. In Fig. 8, we provide the equivalent set of results for all the tasks with $K = 50$ and in Fig. 9 with $K = 200$, half and double the size, respectively.

In view of these results, we can conclude that the results are robust to the choice of this parameter, since we can derive the same conclusions for all values of $K \in \{50, 100, 200\}$: MF-GFN obtains the best trade-off between mean energies and diversity, all other GFlowNet-based methods are able to discover diverse samples and using PPO as the sampler discovers high-scoring samples but strongly lacks diversity.

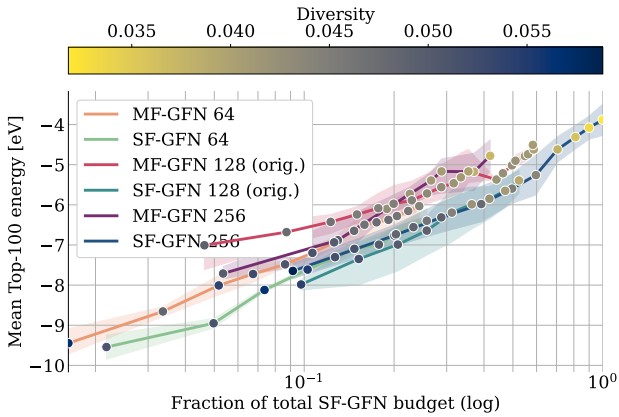

Figure 7: Impact of the acquisition size (64/128/256) on the small molecules (IP) task.

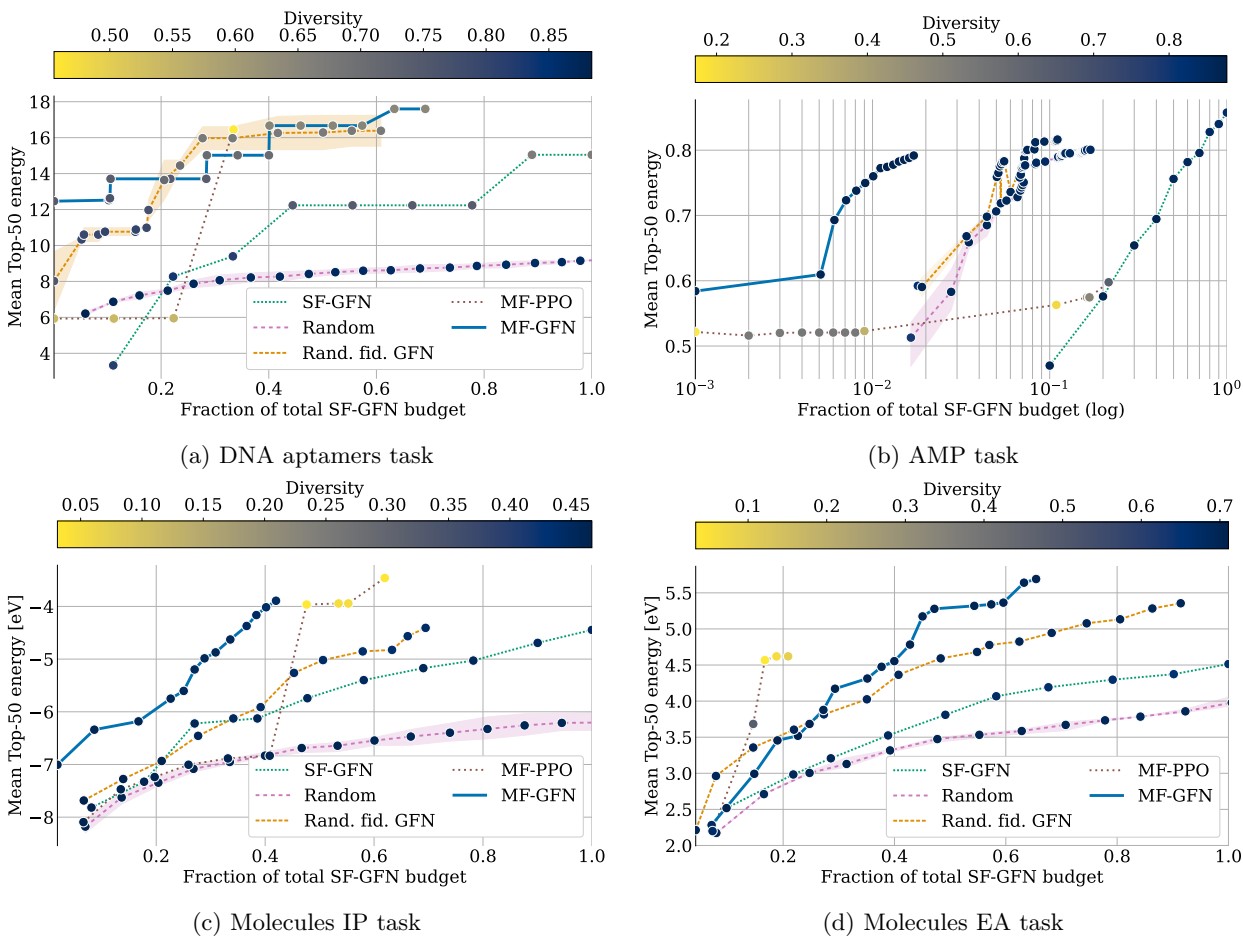

Figure 8: Results as in the original figures, but with $K = 50$, instead of $K = 100$.

## E.5 Visualisation of Sampled Candidates

Given that MF-GFN conducts a cost-aware search with the help of the multi-fidelity acquisition function, our expectation is that the algorithm will selectively query the less costly oracles for input space exploration and will query the more expensive oracles on high-reward candidates. To substantiate this hypothesis, we

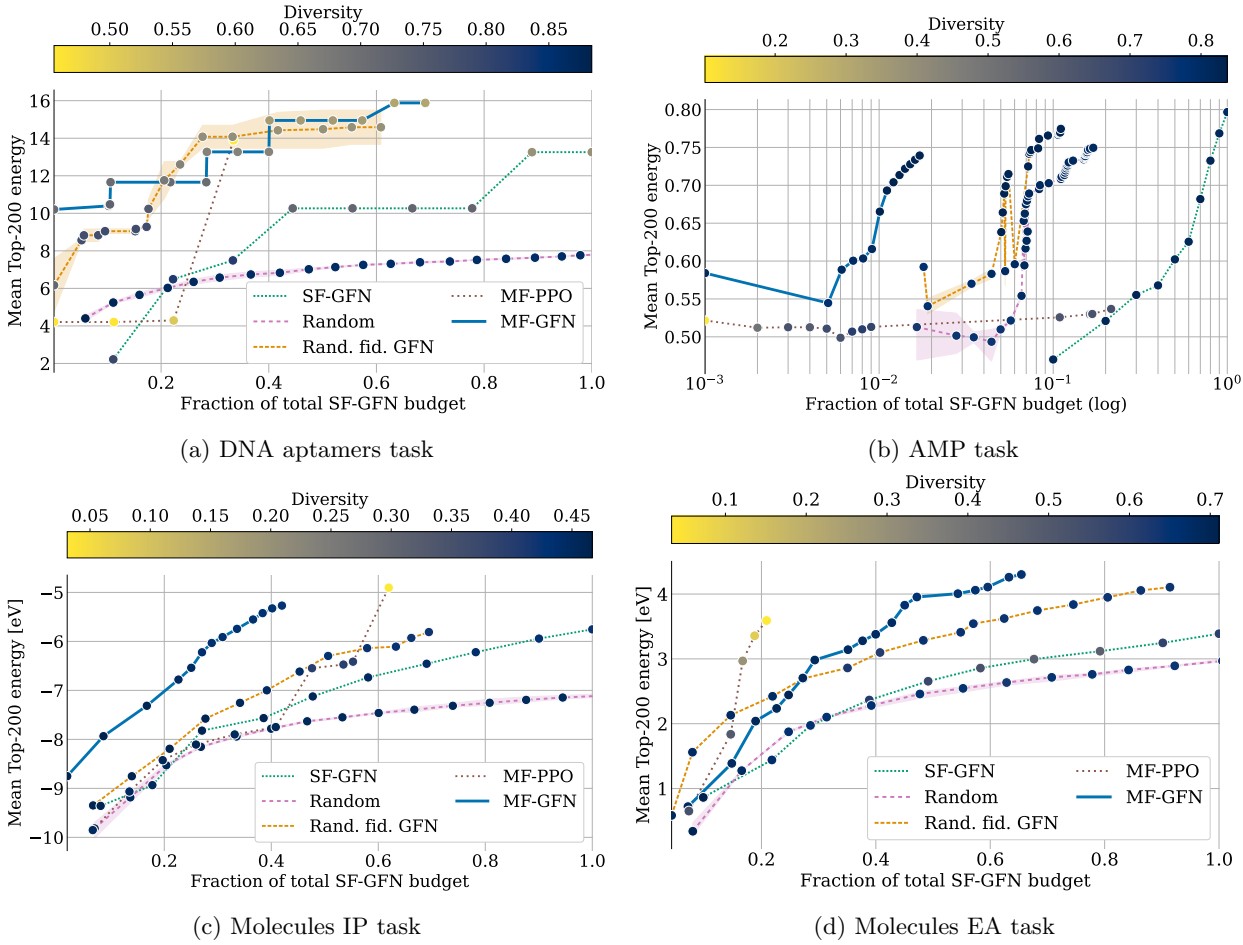

(a) DNA aptamers task

(b) AMP task

(c) Molecules IP task

(d) Molecules EA task

Figure 9: Results as in the original figures, but with $K = 200$, instead of $K = 100$.

provide a two-dimensional visualisation (Fig. 10) of the sampled candidates after expending the allocated budget in the synthetic Branin task (Appendix C.4.1).

Overall, we observe that the lower-fidelity oracles are queried extensively for exploration of a relatively large portion of the sample space, while the highest-fidelity oracle tends to be queried near the modes of the objective function.

## E.6   GFlowNet with random fidelities proportional to inverse of cost

In Section 4.2, we describe the baselines used as comparisons against MF-GFN. The set of baselines includes a slight but relevant modification of MF-GFN, where the policy to select the fidelity of each candidate is not learnt by the GFlowNet but sampled randomly from a uniform distribution (Random fid. GFN). This comparison has allowed us to conclude that the GFlowNet variant that we have introduced, which learns to sample tuples of candidate and fidelity, obtains better results.

A reasonable alternative to sampling the fidelity from a uniform distribution is to sample the fidelity from a distribution proportional to the inverse of the cost, that is $p(m|x) \propto \frac{1}{\lambda_m}$. In order to verify whether this is a significantly stronger baseline than Random fid. GFN, the experiments in Fig. 11 show a comparison on the Hartmann task. As a main conclusion, the two baselines with random fidelities perform very similarly and not better than MF-GFN. A limitation of this analysis is that the simplicity of the Hartmann task does not allow MF-GFN to obtain a large advantage with respect to these baselines.

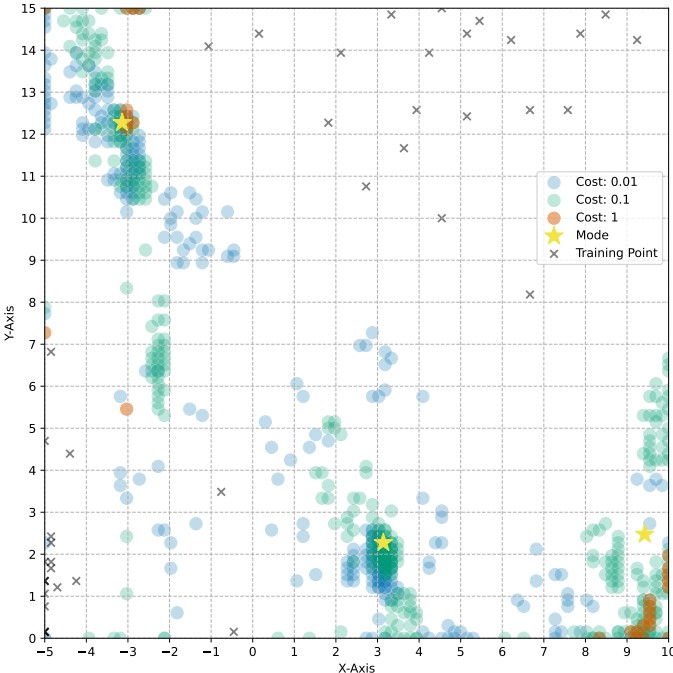

Figure 10: We present a visualisation of the sampled candidates $(x, m)$ in the synthetic Branin task (Appendix C.4.1). The domain of Branin is defined in $[-5, 10] \times [0, 15]$. Each round marker, identified by grid-specific coordinates, represents a sampled candidate, $x$. The markers are colour-coded based on the oracle the candidate is to be evaluated with, $m$. Our observation reveals that the lower fidelity oracles (with costs of 0.01 and 0.1) are primarily used for exploration across the input domain, while evaluations using the high-fidelity oracle (cost=1) are predominantly concentrated near the modes (denoted by the star marker). Note that the training points were intentionally chosen to exclude the modes.

A closer analysis of the results indicates that sampling the fidelities proportionally to $\frac{1}{\lambda_m}$ is strongly biased towards selecting the lowest fidelity. This may result in the stagnation of the improvement as the active learning algorithm progresses, since most often the lowest fidelity is selected, preventing the algorithm to get closer to the modes of the objective function. Given the positive results of MF-GFN in the rest of the experiments and its nearly negligible overhead with respect to these baselines, we conclude that learning the fidelity rather than selecting it at random is a favourable choice.

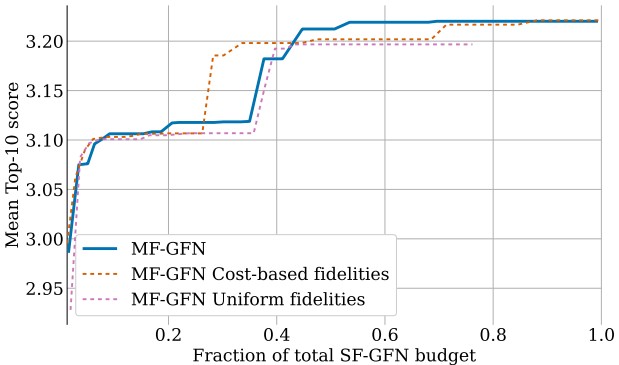

Figure 11: Evaluation of the performance of the MF-GFN baseline which samples the fidelity $m$ from a distribution proportional to the inverse of the oracle cost $\frac{1}{\lambda_m}$. These experiments are on the Hartmann task. We observe that this alternative baseline performs similarly to the one included in the main results section.

