# OpenReview forum: "Multi-Fidelity Active Learning with GFlowNets"
_TMLR — Accepted by TMLR_

### Review · Reviewer_teMv · 2024-04-21

**Summary Of Contributions:**

The paper studies the problem of active learning with multi-fidelity oracles. Particularly, the paper proposes to utilize the GFlowNet trained on the samples and the acquisition functions to learn the policy to generate new samples. The paper shows that on both synthetic tasks and real-world tasks for scientific discoveries, the proposed method is able to generate samples with high scores and diversities.

**Audience:**

Yes

**Claims And Evidence:**

Yes

**Requested Changes:**

1. The diversity metric is a bit vague. I suggest adding detailed equations for such metrics. For example, in the DNA task, it is mentioned that "Diversity is computed as one minus the mean pairwise sequence identity among a set of DNA sequences." However, if I understand it correctly, such a metric is actually the negative of diversity.

2. The notation for augmenting the state space of GFlowNets with the multi-fidelity is also confusing. It seems to me that if a new dimension of multi-fidelity is added, the space should be S cross M instead of the union.

**Strengths And Weaknesses:**

Strengths:
1. The paper extends the GFlowNets to the multi-fidelity setting so that the network is able to utilize oracles of various costs/accuracies to get similar results with greatly reduced overall computational costs.

2. The paper applies the proposed method to a variety of real-world scientific discovery tasks and achieves both good performance and diversity.

Weaknesses:
1. The paper does not have significant technical novelty as the major contribution is to extend the existing GFlowNets to the multi-fidelity scenario.
2. Why GFlowNets can help generate diverse samples is not presented very clearly in the paper, especially after we introduce the multi-fidelity factor.

---

> ### Author Response · Authors · 2024-05-18
>
> First of all, thank you for reviewing our submission. We are confident that the updates to our manuscript motivated by your comments and suggestions have improved the paper. Below, we will summarise these changes and respond to your questions and comments. Note that the update manuscript reflects in orange colour the recent changes to facilitate their identification, and we have included a few margin notes to further clarify the changes.
>
> ### Weaknesses
>
> > The paper does not have significant technical novelty as the major contribution is to extend the existing GFlowNets to the multi-fidelity scenario.
>
> In this regard, we would like to first argue that the extension of the applicability of GFlowNet for multi-fidelity active learning is not a trivial contribution, since one could think of multiple options to do so and we have proposed one that is simple yet effective. Our methodological contribution by which we adapt the sampling mechanism to include the selection of the fidelity for each sample aims to strike a balance between the cost and the accuracy of the evaluation. The active learning setup used to discover diverse, high reward samples has four main components: the surrogate model, acquisition function, the GFlowNet, and the oracles. Notably, we've adapted each of these components to fit the multi-fidelity setting after thorough experimentation. Additionally, we have achieved promising outcomes in the context of small molecules experimentation—a setting that, to the best of our knowledge, has not previously been explored within the multi-fidelity framework. Furthermore, we have thoroughly evaluated the contributions of various components of the algorithm through ablation studies. We strongly believe that our work makes a significant contribution to the multi-fidelity active learning literature, enabling settings which were previously intractable.
>
> > Why GFlowNets can help generate diverse samples is not presented very clearly in the paper, especially after we introduce the multi-fidelity factor.
>
> We have extended sections 3.1 and 3.3 with explanations of why the proposed active learning algorithms with GFlowNet as a generative model that samples proportionally to the acquisition function is able to discover diverse candidates, unlike standard Bayesian optimisation (BO) methods.
>
> In a nutshell, BO *optimises* the acquisition function, resulting in one optimal candidate, while GFlowNet *samples* from the acquisition function, resulting in multiple candidates.
>
> ### Requested changes
>
> > The diversity metric is a bit vague. I suggest adding detailed equations for such metrics.
>
> We agree that the descriptions in the original submission lacked details. We have extended Appendix D with details about not only the diversity metrics but also about the rest of metrics. In particular, about the diversity metrics, we have included the equation of algorithm to compute it for each task (DNA, antimicrobial peptides and molecules).
>
> > The notation for augmenting the state space of GFlowNets with the multi-fidelity is also confusing. It seems to me that if a new dimension of multi-fidelity is added, the space should be S cross M instead of the union.
>
> Indeed, this was mistake in the original submission, which we have corrected in the updated manuscript. Thank you for spotting this error.

---

### Review · Reviewer_JQdA · 2024-04-23

**Summary Of Contributions:**

The authors consider applying GFlowNets in the context of multi-fidelity active learning. This allows amortized, diverse sample generation in structured spaces. Nice results are presented on "toy" and real scientific applications including DNA aptamer design.

**Audience:**

Yes

**Claims And Evidence:**

Yes

**Requested Changes:**

Clarifications and improved presentations. I think the results/technical contribution is sufficient for TMLR already.

**Strengths And Weaknesses:**

The paper is generally clear and well written. I usually find myself advocating for moving detail to the appendix when reviewing (particularly for TMLR where there is no strict page limit), but in this case I actually think too little detail is given in the main text so one is always having to go to the appendix. In particular I think the downsampling kernel and key equations for MES should be in the main text since these are so fundamental to the approach. I would also like more intuition explanation for both of these. "downsampling kernel" does not appear as a term in the cited Wu et al 2019 paper. Is this a choie of the authors? Why is this functional form chosen? For MES it is unclear where the IGApprox expression comes from. I think m is being reused to mean something other than the fidelity level which adds to the confusion. Fig 10 is also interesting enough that I would consider moving it to the main paper.

The work appears technically sound. It's a bit unclear to me how the method biases towards cheaper oracles - is this through the downsampling kernel or built into alpha (eqn 2 in the appendix). In either case, what is the motivation for these functional forms?

Other minor comments/questions:
- p3 there is a space after "drug discovery"
- for eqn 1 (GFlowNet objective), is this summed over all trajectories x?
- for algo 1 is N not also an input?
- for the summation at the end of the algo the superscript should not include i=
- is the downsampling kernel assuming similar m => similar accuracy? is this a good assumption?
- how is m scaled for the downsampling kernel? presumably in [0,1] but this isn't specified
- for appendix B.2 shouldn't the LHS be alpha(x) ?
- next eqn D should be mathcal
- next eqn: sample from f* means sample f then take argmax?
- formula for correlation is missing sqrts in the denominator
- C.5.1 "explained variance" = PERCENT variance?
- fig 10: why are no Cost=1 points near the (3,2) mode?

---

> ### Author Response · Authors · 2024-05-18
> **Part 1/3**
>
> Thank you, first of all, for reviewing our submission. It is encouraging to read that you found our work technically sound and the manuscript well written. We appreciate the multiple suggestions for improvement, including the identification of typos or small errors. Note that the update manuscript reflects in orange colour the recent changes to facilitate their identification, and we have included a few margin notes to further clarify the changes. Let us address all your questions and comments below.
>
> ### Moving details from appendix to main text
>
> Especially regarding the details about the surrogate model, the kernels and the acquisition function, we agree and we appreciate the suggestion.
>
> In the updated manuscript, we have included the expressions of the GP kernels in the main text, as well as intuitions about these choices and pointers to the relevant works in the literature. Furthermore, we have partially rewritten appendix B.1 and B.2 in order to improve the clarity of the exposition, solidify the notation and make it consistent with the rest of the manuscript.
>
> We have also included the expression of the multi-fidelity max-value entropy search (MF-MES) acquisition function in the main text, since we agree the acquisition function is a fundamental component of the algorithm. Additionally, we have partially rewritten appendix B.3 to improve the presentation, notation and connection to the main text.
>
> We are very interested to know whether these changes address your concerns and whether you think that additional details should be included in the main text.
>
> ### Multi-fidelity GP Kernel
>
> We agree that details about the multi-fidelity GP kernel were not clear in the original version. While checking the details of the kernels used in our experiments to write this response and clarify the details in the updated manuscript, we have realised that the experiments reported in the paper were in fact performed with a kernel different than the one reported in the original manuscript. Let us clarify this confusion:
>
> - The kernel we actually used in the reported experiments is the linear truncated kernel as [implemented in BoTorch](https://botorch.org/api/_modules/botorch/models/kernels/linear_truncated_fidelity.html) and as described as well by Mikkola et al. (2023) in their appendix F.1.
> - The kernel we had initially reported is the downsampling kernel, also [implemented in BoTorch](https://botorch.org/api/models.html#botorch.models.kernels.downsampling.DownsamplingKernel) and described by Mikkola et al. (2023) in appendix F.1.
> - The reason of the confusion is that the use of one kernel or the other depends on one boolean argument of [BoTorch’s class SingleTaskMultiFidelityGP](https://botorch.org/api/_modules/botorch/models/gp_regression_fidelity.html#SingleTaskMultiFidelityGP) and the documentation is rather confusing.
>
> To add further confusion, BoTorch uses the term downsampling kernel differently than Mikkola et al. (2023). In any case, in Section 3.2 we have included the explicit expression of the linear truncated kernel we actually used in the experiments, together with the relevant references in the literature that have used it. We have also rewritten Appendix B.1 to provide the necessary background on Gaussian Processes and kernels.
>
> ### Information Gain approximation of MES
>
> The expression of $IG^{Approx}$ is borrowed directly from the GIBBON paper (Moss et al, 2021), Section 3.3, Theorem 3. You were right in your review that $m$ was confusingly reused in our description of the GIBBON approximation in Appendix B.3. Therefore, we have rewritten the section to solidify the notation and clarify the section.

---

> > ### Author Response · Authors · 2024-05-18
> > **Part 2/3**
> >
> > ### Suggestion to move Figure 10 from appendix to main text
> >
> > While we agree that Figure 10 is very interesting, we have decided in this case to keep the figure in the appendix for the following reasons. For context, this figure corresponds to the experiments on a 2D grid with the Branin function as oracle. Importantly, these experiments are not included in the main body of the paper and therefore, in order to include Figure 10, we would need to also include sufficient context about this set of experiments. However, we had decided to keep these experiments in the appendices because the main contribution of our paper is a multi-fidelity active learning algorithm with GFlowNet as the sampler, which allows for the exploration of large, high-dimensional and structured spaces, unlike most of the Bayesian optimisation literature. This is why the main experiments are on DNA sequences, antimicrobial peptides and small molecules. It is within this context that we include the experiments with the Branin and Hartmann objective functions on grid spaces only in the appendix and for completeness. For these reasons, we would prefer to keep Figure 10 in the appendix. This said, we are open to hear other reasons in favour of moving it to the main text.
> >
> > As a compromise, we have updated Section 4.4 with an explicit mention of the experiments with the Branin and Hartmann functions, as well as of Figure 10.
> >
> > ### How the algorithm takes advantage of lower fidelities
> >
> > In your review, you asked further clarification about “how the method biases towards cheaper oracles”. This is a good question because this is the core objective of a multi-fidelity active learning algorithm: to take advantage of the availability of lower fidelity, lower cost oracles.
> >
> > While most components of the algorithm are designed towards this end, the multi-fidelity acquisition function plays a crucial role, as it quantifies the worth of annotating a candidate with a specific oracle. Let us recall the analytical expression of the multi-fidelity max-value entropy search acquisition function $\alpha(x, m)$ we use in our experiments:
> >
> > $$
> > \alpha (x, m) = \frac{1}{\lambda_{m}} I(f_M^\star; f_{m}(x) | \mathcal{D}_{j}),
> > $$
> >
> > where $x$ is a candidate, $m$ is the fidelity of the oracle, $f_m$ the oracle function, $\lambda_m$ the cost of oracle $f_m$, $f_M^\star$ the optimum of the highest-fidelity oracle and $\mathcal{D}_j$ the data set at iteration $j$. $I(A, B)$ indicates the mutual information between random variables $A$ and $B$.
> >
> > Therefore, we can identify two main components in the acquisition function: One, the mutual information in the expression quantifies the information between evaluating candidate $x$ with oracle $m$ and the optimum of the objective function (highest fidelity oracle, $f_M^{\star}$). In other words, pairs $(x, m)$ of candidate and oracle with large amount of information about the optimum will result in larger values of the acquisition function and hence higher probability of being selected. However, the remaining factor in the expression of the acquisition function, $\frac{1}{\lambda_m}$ plays a critical role in “biasing” the acquisition function towards cheaper oracles. The lower the cost of the oracle $\lambda_m$, the higher the value acquisition function.
> >
> > Finally, besides the important role of the acquisition function, the multi-fidelity GFlowNet we introduced in our manuscript is designed to *sample* diverse candidates with high values of the acquisition function. In particular, after training the GFlowNet in an active learning round, it will have learnt a sampling policy $\pi_{\theta}(x, m)$ that generates candidates-oracle pairs $(x, m)$ (approximately) proportionally to the acquisition function $\alpha(x, m)$, since it is used as the reward of the GFlowNet. In other words, if multiple pairs of candidate and oracle achieve high values of the acquisition function, they will be generated by the GFlowNet with high probability. And these candidate-oracle pairs are likely to be from cheaper oracles (low $\lambda_m$) and with high predicted value of the oracle.
> >
> > We are unsure about what exactly you meant with the question about the “functional forms”. We hope that the above description sheds light on your questions. We remain open to further discussion, in any case.

---

> > > ### Author Response · Authors · 2024-05-18
> > > **Part 3/3**
> > >
> > > ### Other questions
> > >
> > > > for eqn 1 (GFlowNet objective), is this summed over all trajectories x?
> > >
> > > The expression in Equation 1 of our manuscript refers to one single trajectory $\tau$. In practice, a the learnable parameters of a GFlowNet are updated each iteration by computing the gradient of the average loss over a mini-batch of trajectories.
> > >
> > > > is the downsampling kernel assuming similar m => similar accuracy? is this a good assumption?
> > >
> > > Yes, that’s right (for both the linear truncated kernel and the previously reported downsampling kernel). Assuming that similar fidelity levels yield similar accuracy seems a reasonable assumption to us and it is indeed a modelling assumption made widely in the multi-fidelity Bayesian optimisation literature.
> > >
> > > > how is m scaled for the downsampling kernel? presumably in [0,1] but this isn't specified
> > >
> > > First, as discussed above, we mistakenly reported using the downsampling kernel but we actually used the linear truncated kernel. Now, to the question of how the fidelities are scaled for the kernel, they are indeed, as you said,
> > >
> > > > C.5.1 "explained variance" = PERCENT variance?
> > >
> > > The "explained variance" is a metric quantifying how much of the variation in a target variable is captured by the oracle’s predictions. A higher explained variance indicates a better fit, meaning our model accounts for a larger proportion of the observed fluctuations in the data. We computed explained variance using the implementation in Python's scikit-learn library. The exact mathematical function is
> > >
> > > $$
> > > \text{explained variance}(y_{\text{true}}, y_{\text{pred}}) = 1 - \frac{\text{Var}(y_{\text{true}} - y_{\text{pred}})}{\text{Var}(y_{\text{true}})}
> > > $$
> > >
> > > If the question is whether it can be interpreted as a percentage, that is correct, since the explained variance is in [0, 1], with 1 indicating a perfect fit.
> > >
> > > > fig 10: why are no Cost=1 points near the (3,2) mode?
> > >
> > > We don’t have a strong hypothesis about this. What we can say is that we do not expect the algorithm to always select the near-optima points with the highest-fidelity. Once a near-optimum point has been modelled with an annotation from a lower-fidelity oracle, then the value of the acquisition function on that point with the highest-fidelity oracle would decrease because the information gain would be lower.
> > >
> > > ---
> > >
> > > All other minor comments referring to typos or small errors have been corrected in the updated manuscript.

---

### Review · Reviewer_as9b · 2024-04-26

**Summary Of Contributions:**

This paper addresses the problem of black-box optimization in a setting where finding a diverse set of potential solutions is important. Unlike prior work, it considers the multi-fidelity setting, where multiple approximations of the target function at different costs are available for querying. The paper presents an approach to this problem through the use of GFlowNets, deep kernel learning with multi-fidelity kernels, multi-fidelity acquisition functions, and an iterative algorithm on top of these. The approach is evaluated on four biochemical tasks as well as two standard synthetic optimization problems, showing strong results with small budgets and high diversity of the obtained batch.

**Audience:**

Yes

**Broader Impact Concerns:**

No concerns

**Claims And Evidence:**

Yes

**Requested Changes:**

Overall, I like the paper, and I think it should be accepted with some minor revisions. The points mentioned in the weaknesses should, in my opinion, at least be discussed in the paper. Experiments would be better, especially for the first point, but it depends on the effort needed to include them.

**Questions:**
- Why do many curves stop before reaching the maximum budget? Is the computation too slow? Why do some have error bars and some don't? What do the error bars mean? (This should be discussed and/or improved in the paper.)
- Section 3.2, last paragraph: "MES [...] makes random sampling a strong baseline": What is meant by that? How is random sampling used there? Like in successive halving? In particular, if an acquisition function can make random sampling better, it must mean that random sampling uses the acquisition function somehow? (Please provide a bit more detail in the paper.)

**Minor Comments:** (should be easy to resolve, but they are not critical for acceptance)
- Section 3.2, paragraph "Surrogate Model", last sentence: Can the authors briefly explain what is a "linear downsampling kernel over the fidelity index" (or refer to the Appendix)? (It could be mentioned that the used kernel is a product kernel of a deep kernel for the $x$ part and a fixed kernel for the $m$ part.)

- Algorithm 1: In the second line, shouldn't it be $f_{m_i}$ instead of $f_m$?
- Algorithm 1: What is $\beta$?
- Algorithm 1: At "Result": How is "Diversity" computed? (Maybe it should not be returned but computed outside of the algorithm to avoid confusion.) How is Top-K(D) computed? Based on the surrogate for the highest fidelity? Maybe the "Result" line should be moved to the end of the algorithm as a "return" statement?

- Section B.1: In the first align/eqnarray, you used $x$ but the text above suggests that you wanted to use $z$ for these equations?
- Section B.1: Equation for $K_n(x_1, x_2)$ has an extra ( and $x1:n$ instead of $x_{1:n}$.

- Table 4: Should be $(\beta_1, \beta_2)$ instead of $(\beta 1, \beta 2)$.

**Strengths And Weaknesses:**

**Strengths:**
- The paper is well-written.
- The presented method is evaluated on multiple challenging and somewhat realistic problems.
- The presented method achieves very good results in the benchmarks. (I am avoiding the term "state-of-the-art" here because the benchmarks do not appear to be well-established.)
- Multiple ablations are included.
- Code is provided.

**Weaknesses:**
- I am not fully convinced about the way that the selection of the fidelity is handled, because while the acquisition function incorporates the cost of the fidelities, using a fixed fidelity kernel might not allow the model to learn the actual utility of the different fidelity levels. I would be interested in the following experiments: (a) Random fidelity GFN where the fidelity distribution is not uniform but proportional to $1/\lambda_m$. This way, the same cost is spent on all fidelities in expectation. (b) MF-GFN vs. SF-GFN in a two-fidelity setting where the cheaper fidelity is not useful at all (e.g. constant or pure noise). Does MF-GFN learn to query this fidelity less over time? (If it turns out that MF-GFN cannot learn the utility of a fidelity, it might be possible to fix this by including $m$ as an input to the surrogate model instead of including it through a separate kernel, but I expect that this might require some non-trivial and problem-dependent architectural changes. Therefore, I do not expect the authors to actually do this.)
- Algorithm 1 selects an unspecified number $N \gg B$ of tuples and then picks the top $B$ out of them. As in pool-based active learning, making $N$ too large could lead to the top $B$ samples being near-identical. I did not find a discussion on how to choose $N$ or whether this is a sensitive hyperparameter in practice. It could be interesting to look at diversification strategies, such as a simple stochastic acquisition (https://arxiv.org/abs/2106.12059), although this is already what the policy $\pi_\theta$ should have learned to do. I leave the decision whether to do additional experiments on this to the authors, as I do not know how expensive they would be.

---

> ### Author Response · Authors · 2024-05-18
> **Part 1/2**
>
> Thank you for the thorough review of our paper. We are encouraged by the positive comments about our work, and we greatly appreciate the multiple suggestions for improvement, including experiments and minor changes. We have worked to incorporate these suggestions and correct the errors. We will address all your comments and questions below.
>
> ### Selection of the fidelity
>
> In your review, you expressed concerns about how the selection of the fidelity is handled by the algorithm. Specifically, that “using a fixed fidelity kernel might not allow the model to learn the actual utility of the different fidelity levels”.
>
> The multi-fidelity surrogate used in our paper is inspired from the multi-fidelity Bayesian optimisation literature. In particular, we adopt the linear truncated kernel as [implemented in BoTorch](https://botorch.org/api/_modules/botorch/models/kernels/linear_truncated_fidelity.html), which had been used, for instance, by Mikkola et al. (2023) and is described in their appendix F.1. Please note that in the original submission we had mistakenly reported the use of the downsampling kernel, closely related, but during the rebuttal period we have found out that we had used indeed the linear truncated kernel. The mistake was due to the confusion in reading the documentation of [BoTorch’s class SingleTaskMultiFidelityGP](https://botorch.org/api/_modules/botorch/models/gp_regression_fidelity.html#SingleTaskMultiFidelityGP), where the use of one kernel or the other depend on one boolean argument. We have updated Section 3.2 as well as Appendix B with the corrected and clarified information about the kernels used.
>
> Besides this note, an important difference with the work by Mikkola et al. is that we adopted a deep kernel learning approach with variational Gaussian Processes. In principle, the use of this multi-fidelity kernel should allow the surrogate to learn the posterior of each oracle $m$ $p(f_m(x) | x, m, \mathcal{D})$. Why do you argue that the model would not lean the utility of the fidelity levels? In your review, you suggest to include $m$ “as an input to the surrogate model instead of including it through a separate kernel”. Since the kernel is part of the surrogate model, the fidelity level is effectively one of its inputs. Did you mean that the fidelity could be an input to the neural network $g_{\omega}$ that learns an embedding of the inputs $x$?
>
> Regarding the utility of a fidelity, it is important to note that the actual utility used by the algorithm is quantified by the acquisition function, which estimates the mutual information between an oracle $f_m$ and the maximum value of the highest-fidelity oracle $f_M^\star$, divided by the oracle cost $\lambda_m$.
>
> ### Suggested experiments
>
> In your review, you suggest two experiments to assess the ability of the algorithm to capture the utility of each fidelity.
>
> a) Random fidelity GFN where the fidelity distribution is not uniform but proportional to $\frac{1}{\lambda_m}$.
>
> We appreciate this suggestion because we agree that this would be a stronger baseline than the version with uniformly sampled fidelities that we used in our original results. Therefore, the comparison of this baseline with MF-GFN should give a better indication of the actual contribution of learning to sample the fidelity index. Unfortunately, we have not had time to prepare and run these experiments, but we are very much willing to run them and report back as soon as we have the results.
>
> b) MF-GFN vs. SF-GFN in a two-fidelity setting where the cheaper fidelity is not useful at all
>
> This one is also interesting and it is easy to show that, in theory, the cheaper oracle would not be selected because the acquisition function should be zero for pairs $(x, m)$ where $m$ corresponds to the useless oracle. Let us recall the expression of the acquisition function:
>
>
>
> $$
> \alpha(x, m) = I(f_M^\star;f_m(x)|\mathcal{D}_j)
> ,
> $$
>
> which can be rewritten in terms of entropy and conditional entropy:
>
> $$
> I(f_M^\star;f_m(x)|\mathcal{D}_j)=H(f_m(x)|\mathcal{D}_j) - \mathbb{E}_{f^\star}[H(f_m(x)|f_M^\star, \mathcal{D}_j)|\mathcal{D}_j].
> $$
>
> If an oracle $f_m$ is constant or noise, then that would make it independent of $f_M^\star$. Under the assumption that $f_m \perp f_M^\star$
>
> $$
> I(f_M^\star;f_m(x)|\mathcal{D}_j) = H(f_m(x)|\mathcal{D}_{j}) - H(f_m(x)|\mathcal{D}_{j}) = 0.
> $$
>
> As can be seen, this theoretical property is a feature of the acquisition function, rather than the multi-fidelity GFlowNet.
>
> In practice, unfortunately, the mutual information is estimated via the GIBBON approximation (see the rewritten Appendix B.3) and the surrogate model is also fit via variational GPs. This is likely to result in inaccurate estimations of the acquisition function under a noisy or constant oracle.

---

> > ### Author Response · Authors · 2024-05-18
> > **Part 2/2**
> >
> > We have performed two empirical evaluations of this suggested setting:
> >
> > - We have run the MF-GFN algorithm on both the Hartmann function and the DNA task, by setting all oracles but the highest-fidelity oracles to $f_0(x) = 0, \forall x$. We have observed that the algorithm fails to capture the useless utility of the cheaper oracles: while in some active learning rounds the samples are all assigned the highest-fidelity oracle, this is not consistent and even in some rounds all samples are assigned the constant oracle.
> > - In light of the above results, we wondered whether the issue would be in the estimation of the acquisition or in the sampling by the GFlowNet. Therefore, we have conducted isolated experiments where we estimate the acquisition function as in our experiments on the entire domain of the Branin function, which can be easily visualised and analysed. We have observed that the estimated value of the acquisition function is always non-zero and not perfectly homogeneous across the domain, contrary to the theoretical values. Therefore, depending on the cost of the constant oracle, the value of the acquisition function might be even higher than the acquisition function evaluated for the highest fidelity.
> >
> > We thus conclude that the overall performance of the algorithm is impacted by the approximations made in the surrogate model and in the estimation of the acquisition function, as well as by other components, which is not surprising.
> >
> > ### Impact of $N$, number of samples from GFlowNet
> >
> > Algorithm 1 indicates indeed that $N \gg B$ and in Appendix C.1 it is specified that $N = 5 \times B$. We agree with your suggestion that more sophisticated batching strategies could provide additional benefits to the algorithm and for this reason we have added this idea and the mention of stochastic batch acquisition to the future work section.
> >
> > Nonetheless, we would like to note that given the query synthesis nature of our active learning modality, with combinatorially large design spaces, and the use of GFlowNets, it is likely that the batch of selected candidates preserves a high degree of diversity, despite using an even larger $N$.
> >
> > ### Other questions
> >
> > > Why do many curves stop before reaching the maximum budget? Is the computation too slow?
> >
> > Some curves stop before reaching the maximum budget due to an “early stopping” mechanism introduced to avoid wasting computation when the average scores stopped improving. We agree that this is confusing and therefore we intend to re-run the experiments without early stopping and update the figures by the next revision.
> >
> > > Why do some have error bars and some don't? What do the error bars mean?
> >
> > The error bars indicate the 95 % confidence intervals estimated via bootstrapping. We have included this information in Section 4.1 in the updated manuscript. The missing “error bars” are due to errors in the storage of the some results in the DNA and AMP experiments. We will have to repeat these experiments and update the figures in the next revision. Thank you for noticing this issue.
> >
> > > "MES [...] makes random sampling a strong baseline": What is meant by that? How is random sampling used there? Like in successive halving? In particular, if an acquisition function can make random sampling better, it must mean that random sampling uses the acquisition function somehow?
> >
> > First of all, let us clarify that our “Random” baseline does use the acquisition function, as noted in the description of the baseline in Section 4.2: “Quasi-random approach where both candidates and fidelities are randomly sampled. We query the top (x, m) pairs according to the acquisition function.” (We have updated the manuscript to indicate the the random distribution is uniform.)
> >
> > In other words, this baseline samples $N$ candidates randomly (instead of training a GFlowNet), scores them with the acquisition function and then selects the top $B$. This is why this baseline is “strong”. Given the combinatorially large design space, simply sampling $B$ candidates at random would result in a really weak baseline.
> >
> > In order to avoid any confusion, we have removed the comment about the strong baseline in the description of the MES acquisition function, and added a note in the description of the baseline. We hope that clarifies the matter.
> >
> > > What is $\beta$?
> >
> > The temperature parameter that controls the *peakiness* of the reward distribution, which offers the possibility of sampling high-reward samples (high acquisition) with higher probability. This is detailed in Appendix C.3.
> >
> > ---
> >
> > All remaining minor comments have been addressed in the updated manuscript.

---

> > > ### Comment · Reviewer_as9b · 2024-05-21
> > >
> > > I'll be able to respond in more detail after the NeurIPS deadline, but here is a bit of background on my concerns regarding the multi-fidelity: I know from standard active learning that for GPs / kernel regression with fixed kernels, typical acquisition functions try to cover the whole input space more or less uniformly, irrespective of which labels are observed. On the other hand, when the kernel / features are learned by a neural net, the acquisition can focus more on difficult regions of the space because the NN features capture where these regions are. When we move from MSE-based active learning to black-box optimization, the picture becomes more difficult, but the point remains that with a fixed kernel, the influence between different regions of input space is not really learned. IIRC you have a learned kernel for the design space and a fixed kernel for the fidelity space, from which I would have guessed that the acquisition of fidelities is essentially independent of what labels/scores you observe, and relatively close to the alternative random version that I proposed. It seems that your observations with irrelevant fidelities confirm this to some extent. Because of these considerations, I would find it interesting what happens if you make the fidelity part of the kernel learnable somehow. This is why, indeed, I suggested that the fidelity could be an input to the neural network $g_{\omega}$ that learns an embedding of the inputs $x$, but this might require individual architectural changes, so it is not a very comfortable option. I don't know if it's possible to simply take the current fidelity kernel and make it learnable by applying it to learnable embeddings of m instead of m directly (and training this somehow on all of the fidelities together).
> > >
> > > These are some quick thoughts, and I think investigating this is interesting since the multi-fidelity acquisition is the main novelty of the paper, though it might not be necessary to satisfy TMLR's acceptance criteria.

---

> > > > ### Comment · Reviewer_as9b · 2024-05-24
> > > >
> > > > I forgot to say it in my previous comment, but thank you for your detailed response. To make my previous comment more precise, maybe it would be possible to do deep kernel learning on $K_M$ as well, where the deep part is simply an embedding layer that is initialized as $m \mapsto \lambda_m'$, or something similar.
> > > >
> > > > Regarding your comments on the acquisition function, you write "If an oracle $f_m$ is constant or noise, then that would make it independent of $f_M^\star$". However, the acquisition function is judging whether this independence holds *in the model*, not in reality. If the model is not able to learn that the independence holds, it means that the acquisition will always continue acquiring the useless oracle, as you observed in your experiments. In a GP model, the kernel models the covariance between observations at different fidelities. If the kernel is fixed, you assume a fixed covariance and the model can never learn that the observations are "independent" in reality. (Of course, a kernel method can learn to fit the target function. It just can't learn to adjust its own epistemic uncertainty based on the target labels.)
> > > >
> > > > Overall, I think it would be good to see the experiments on the 1/cost random baseline until the end of the discussion phase next week, and then (perhaps with a discussion of the limitations of the fidelity acquisition) the paper should meet the acceptance criteria for TMLR. If you can provide some experiments on useless fidelities / deep kernel learning for $K_M$, it might make me lean more towards a certification, and might also improve your paper/method, although I understand that there is not much time and it might be a lot of effort to do this especially for the non-synthetic tasks.
> > > >
> > > > Re impact of $N$: Thank you. This is fine for me, although I think $N=5B$ could be mentioned in the main paper.

---

> > > > > ### Author Response · Authors · 2024-05-29
> > > > >
> > > > > Let us first confirm that the algorithm described in our submission, used to obtain the presented results, uses a deep kernel for the design space and a fixed kernel for the fidelities. We appreciate your suggestion of making the fidelity part of the kernel learnable as well, since we share the intuition that this could enhance the performance of the acquisition function. Furthermore, to the best of our knowledge, this would be an original contribution. We are thus very keen to explore this approach. Unfortunately though, implementing these changes and running the experiments will require time that goes beyond the tight deadlines of the review process. We are currently running experiments with MF-GFN with random fidelities sampled proportionally to the cost and we hope to be able to report the results by the end of the week.

---

> > > > > > ### Author Response · Authors · 2024-06-01
> > > > > > **Baseline with random fidelities proportional to inverse of cost**
> > > > > >
> > > > > > As a brief last-minute update, we would like to note that we have updated the manuscript with a small addition: a section in the supplementary material (E.6) with a presentation of results with one of the baselines suggested in your review, namely a random fidelity GFN where the fidelities are sampled proportionally to $$\frac{1}{\lambda_m}$$. we have conducted the experiment on the Hartmann task. As main conclusions, we observe that its performance is very similar to the random fidelity GFN with uniformly sampled fidelities, and worse than MF-GFN. We have observed that the strong bias of this baseline towards the lowest fidelity renders it very slow, requiring many active learning rounds to get closer to the modes of the black-box function.

---

> > > > > > > ### Comment · Reviewer_as9b · 2024-06-10
> > > > > > >
> > > > > > > Thank you for the update!

---

### Author Response · Authors · 2024-05-09
**Extension of rebuttal period**

Dear reviewers,

First, we would like to thank you for your work on reviewing our manuscript. We are currently working on the responses to your questions and comments, as well as incorporating your feedback into the manuscript. However, due to the overlap of the rebuttal period with ICLR, we have requested the Area Chair an extension, which has been accepted. We will submit our responses by May 17th.

Thank you for the understanding

---

### Decision · Action_Editor_zRDt · 2024-06-17

**Recommendation:** Accept as is

**Comment:**

I am pleased to inform you that the recommendation is for your paper to be accepted in TMLR. The work presents a valuable extension of GFlowNets to handle multi-fidelity optimization with promising results across several application domains. The reviewers appreciated the thorough experiments demonstrating the utility of your method. The additional edits during the discussion period have improved the overall presentation and readability --- I want to thank the authors and reviewer as9b, in particular, for making good use of the discussion period and review process.

**Audience:**

Yes, all three reviewers agree that this paper is of interest to the TMLR audience. While reviewers noted that the method's technical novelty is somewhat incremental, combining GFlowNets with multi-fidelity Bayesian optimization techniques, this is of no relevance according to the TMLR acceptance criteria, and the reviewers found the experimental results compelling and felt the contributions would be of interest to researchers working on related problems.

**Claims And Evidence:**

Yes, the reviewers are satisfied with the claims put forth in the paper and the evidence. The experiments demonstrate the utility of the proposed multi-fidelity optimization approach compared to single-fidelity baselines across a range of tasks. Reviewer as9b suggested the inclusion of additional experiments during the discussion period, and the authors added one of the suggested experiments to the appendix and clarifications throughout the paper. Overall, the reviewers found the empirical results to be sound and convincing.